# Leveraging artificial intelligence to identify the psychological factors associated with conspiracy theory beliefs online

Jonas R. Kunst [1] ✉, Aleksander B. Gundersen [1,6], Izabela Krysińska [2,6], Jan Piasecki [3], Tomi Wójtowicz [2], Rafal Rygula [4], Sander van der Linden [5] & Mikolaj Morzy [2]

Given the profound societal impact of conspiracy theories, probing the psychological factors associated with their spread is paramount. Most research lacks large-scale behavioral outcomes, leaving factors related to actual online support for conspiracy theories uncertain. We bridge this gap by combining the psychological self-reports of 2506 Twitter (currently X) users with machine-learning classification of whether the textual data from their 7.7 million social media engagements throughout the pandemic supported six common COVID-19 conspiracy theories. We assess demographic factors, political alignment, factors derived from theory of reasoned action, and individual psychological differences. Here, we show that being older, self-identifying as very left or right on the political spectrum, and believing in false information constitute the most consistent risk factors; denialist tendencies, confidence in one's ability to spot misinformation, and political conservativism are positively associated with support for one conspiracy theory. Combining artificial intelligence analyses of big behavioral data with self-report surveys can effectively identify and validate risk factors for phenomena evident in large-scale online behaviors.

The COVID-19 pandemic, perhaps more than any other crisis since the Second World War, underscored the pervasive and damaging societal impact of conspiracy theories, in this case primarily spread through social media[1–4]. Conspiracy theories can be succinctly defined as beliefs about "the machinations of a small group of powerful people, working in secret, against the common good"[5] (p. 151). Thus, although conspiracy theories evolve continuously and track important societal events, they usually represent shared features (e.g., abuse of power, deception)[6]. Importantly, while these theories are often bereft of credible evidence, they are not necessarily based fully on falsehoods[7].

During the COVID-19 pandemic, conspiracy theories concerning the origins of the virus, its actual existence, severity and prevention,

and the hidden motives surrounding it proliferated. Already by early 2022, estimations indicated that between 150,000 and 350,000 tweets each month conveyed a conspiracy theory, often exceeding the volume of general COVID-19 related content[3]. Consequently, the World Health Organization rapidly declared an "infodemic"—an overwhelming flood of information, often misleading—that ran parallel to the pandemic, exacerbating its already destructive impact[8].

Empirical research swiftly documented the repercussions of these conspiracy theories on global public health. Beliefs in the theories were not only correlated with, but also prospectively predicted decreased compliance with preventive measures and the increased likelihood of resorting to potentially dangerous alternative treatments[1,9,10] (but see

[1]Department of Psychology, University of Oslo, Oslo, Norway. [2]Faculty of Computing and Telecommunications, Poznan University of Technology, Poznan, Poland. [3]Department of Philosophy and Bioethics, Faculty of Health Sciences, Jagiellonian University Medical College, Krakow, Poland. [4]Maj Institute of Pharmacology, Polish Academy of Sciences, Krakow, Poland. [5]Department of Psychology, University of Cambridge, Cambridge, UK. [6]These authors contributed equally: Aleksander B. Gundersen, Izabela Krysińska. ✉e-mail: j.r.kunst@psykologi.uio.no

ref. 11 for a critical discussion of the assumed consequences of conspiracy theories). Furthermore, they negatively impacted intergroup relations, often leading to increased polarization within society and internationally[11,12]. Such detrimental impacts of conspiracy theories accentuate the urgency of discerning psychological factors, which can aid in pinpointing at-risk populations and shaping interventions[13]. Yet, while numerous studies have endeavored to identify these determinants[14], research efforts have been predominantly constrained by an over-reliance on self-reported data and very rarely combined them with large-scale behavioral insights from social media platforms. Conversely, whereas certain studies offer compelling analyses of large-scale online behavior related to conspiracy theories on social media platforms[15], they typically fail to integrate these observations with in-depth psychological insights, which are challenging to gauge accurately without the use of surveys. Owing to the significant inconsistencies between self-reported intentions and actual behaviors[16], our current understanding of the psychological factors that correlate with online support for conspiracy theories as documented in big data remains limited.

Nevertheless, the insights gained from existing research offer valuable guidance for studies that aim to integrate both sources of data. Numerous self-report studies have adopted an approach of investigating individual differences. Notably, narcissism has consistently shown a strong association with conspiracy beliefs, as corroborated by meta-analytic evidence besides others[14,17]. The allure of conspiracy theories for narcissistic individuals may lie in their potential to garner attention, affirm their perceived superiority, and display their unique insights or viewpoints. Moreover, individual differences in denialism and a need for chaos seem to play an important role. Denialism, characterized by the rejection of expert narratives and the tendency to seek unconventional explanations for significant events[18], has been consistently linked to beliefs in COVID-19 conspiracy theories in self-report research[19]. Similarly, the need for chaos, a desire to provoke or exacerbate disorder, has been identified as a key motivator behind the spread of hostile political rumors[20,21].

Individuals' more general political alignment is another crucial factor in understanding their conspiracy theory beliefs. Extensive cross-cultural research[22] indicates that individuals at both ends of the political spectrum are more likely to endorse conspiracy theories[23,24]. Indeed, individuals on both the left and right harbor suspicions that the opposite side engages in conspiracies[25]. Behavioral data, such as user interactions on social media, further support these findings, showing that polarized individuals are more engaged in propagating information related to conspiracy theories[26]. This trend of political extremity underlying conspiracy theory beliefs is particularly pronounced when theories concern powerful entities[27]. Nonetheless, right-leaning individuals show a greater tendency towards conspiracy theories targeting marginalized groups[27]. Certain conservative movements, as for instance exemplified by voting for Trump compared to Biden, still seem to exhibit a stronger overall inclination towards conspiracy beliefs[28].

In examining the support for conspiracy theories, the Theory of Reasoned Action[29] and its recent adaptations to misinformation processing[30] are of relevance. This theory suggests that behavior is influenced by attitudes towards the behavior and normative beliefs. Trust in the authenticity of information on social networks and confidence in identifying false information have been shown to positively correlate, while attitudes towards the importance of validating information have been shown to negatively correlate, with the self-reported spread of misinformation[30,31]. Whereas this research has focused on the broader topic of misinformation, it has replicated the role of overconfidence with misinformation stimuli that often mapped onto conspiracy theories[32]. Moreover, recent evidence suggests a correlation between a general overconfidence in one's abilities and belief in conspiracy theories[33]. However, contrasting findings exist, such as a

study demonstrating that an inoculation intervention enhanced both the ability to discern misinformation (including misinformation related to conspiracy theories) and confidence in judgment, with both constructs being positively correlated[34].

Rather than focusing on identifying related predisposing factors, certain studies have endeavored to directly measure individuals' susceptibility to believing in conspiracy theories or misinformation more broadly through the use of specialized psychometric scales. Arguably most influential has been the conspiracy mentality questionnaire[35], thought to capture a disposition towards conspiratorial viewpoints across various subjects[36]. While the scale was originally conceptualized as distinct from specific conspiracy theory beliefs, it has consistently demonstrated a positive correlation with many of these beliefs[36]. Additionally, a recently developed scale assesses susceptibility to misinformation, by measuring the propensity to accept politically-balanced false information, including conspiratorial content, and the tendency to reject accurate information[37]. Of particular relevance to the present research, higher misinformation susceptibility as indexed by the scale was positively associated with COVID-19 conspiracy beliefs[38].

Although the research landscape on conspiracy theories is developing quickly, it is still marked by diverse and sometimes conflicting perspectives. For instance, while some studies highlight the sizeable explanatory power of personality factors[28], recent data-driven research suggests that distrust in governments (closely related to denialism) and perceptions of powerful others (closely related to conspiracy mentality) may have more explanatory power than personality factors[39]. However, arguably one of the most critical limitations in existing research is its primary reliance on self-report data, which hinders precise identification of individuals and populations behaviorally engaged in or resistant to conspiracy beliefs. This gap underscores the need for research testing the role of theoretically-based factors with large-scale and naturalistic behavioral data to inform targeted interventions.

In response to this need, we leverage a unique dataset, merging self-reported psychological data from 2506 U.S.-based Twitter (currently X) users with machine learning analysis of their 7.7 million engagements on the social media platform spanning most of the duration of the COVID-19 pandemic. We focus on users from the U.S. because the spread of COVID-19 conspiracy theories on the platform was most pronounced in North America[40]. Central to our study is the assessment of an expansive array of psychological variables, previously proposed and tested in mostly self-report studies as reviewed above, to ascertain their possible roles as either risk or protective factors in relation to behavioral support for conspiracy theories online.

We examined a series of hypotheses derived from the literature review presented earlier. Initially, we investigated whether characteristics such as narcissism, denialism, and a need for chaos are correlated with increased support for conspiracy theories in an online context. Subsequently, we evaluated the hypothesis proposing that individuals who self-identify at the very ends of the political spectrum, whether on the far left or far right, are more inclined to support conspiracy theories. Concurrently, we explored whether affiliation with the Republican Party is a predictor of greater conspiracy theory support. Furthermore, grounded in the Theory of Reasoned Action, we probed the hypothesis that trust in the authenticity of information on social networks and confidence in identifying false information are positively associated with endorsing conspiracy theories online. By contrast, attitudes that underscore the importance of verifying information are hypothesized to correlate negatively with such endorsements. Lastly, we examined the hypothesis that participants' conspiracy mentality and susceptibility to misinformation would be positively associated with their support of conspiracy theories online. Beyond the psychological variables that were of core interest to our study, we tested the demographic hypotheses that men, older individuals, and those with

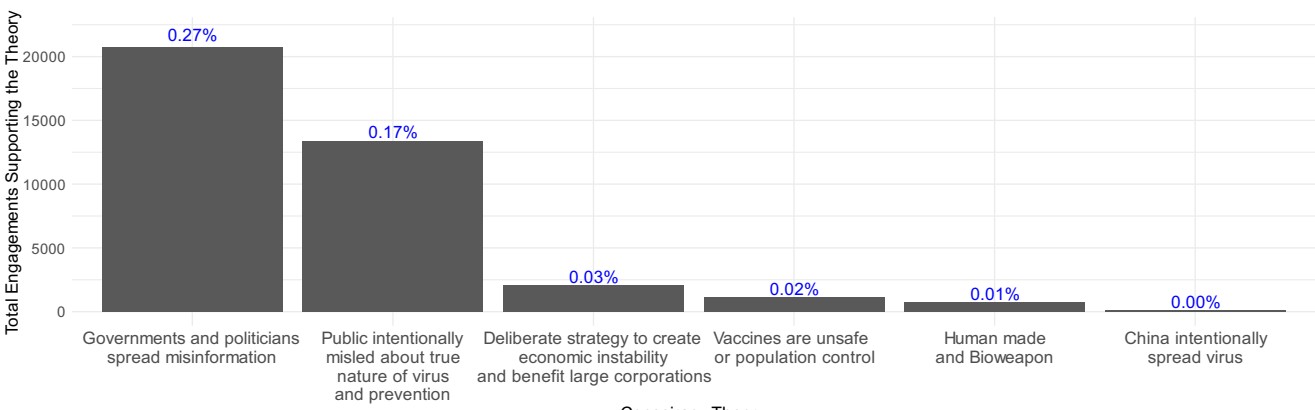

**Fig. 1 | Sum of Twitter (currently X) engagements supporting each of the six conspiracy theories and their percentage of all 7.7 million engagements.** The bars represent the total number of engagements that support each conspiracy theory, while the percentages (%) above the bars indicate the proportion of these engagements relative to the total number of all engagements.

less education exhibit a higher support for conspiracy theories online[22,41–46] (but see ref. [47]).

Our outcome variable constituted the degree to which participants' engagements on the Twitter (currently X) platform in the form of likes, posts, replies, and reposts behaviorally supported six different overarching COVID-19 conspiracy theories derived from previous research[48] (see Fig. 1). Importantly, each of these engagements not only indicates endorsement of conspiracy theories but also ultimately amplifies their spread by prompting algorithms to boost their visibility. As such, our outcome variable, support for conspiracy theories, captures both their endorsement and spread.

In this work, using Natural Language Processing models, we select engagements that are at least minimally semantically related to each conspiracy theory based on their content. Then, we use a commercial-grade model to evaluate whether each engagement supports a given conspiracy theory (see Methods for details). This combination of psychology and AI allows us to scale our research to millions of behavioral data points defined by individual online engagements. For a comprehensive understanding of the data, we scrutinize the relationship between the demographic and psychological variables and behavioral support on Twitter (currently X) for each of the six conspiracy theories. For interested readers, we present the associations between each of the psychological variables and conspiracy theory support estimated separately for the four primary Twitter (currently X) engagement forms (i.e., likes, posts, replies, reposts) in the Supplementary Information. Our findings indicate that the most consistent risk factors are older age, being politically far left or far right, and believing in false information. Additionally, denialist tendencies, confidence in one's ability to spot misinformation, and political conservatism are positively linked with support for one conspiracy theory.

## Results
### Differences in support for conspiracy theories based on their content
As can be expected, the numbers of Twitter (currently X) engagements supporting the selected conspiracy theories were generally low (see Fig. 1). Still, the frequency of support was highest for the theory that governments and politicians intentionally spread misinformation ($n_{Engagements}$ = 20,705), followed by the theory that the public is intentionally being misled about the true nature of the virus and prevention ($n_{Engagements}$ = 13,354), that the purpose of the virus was to create economic instability and benefit large corporations ($n_{Engagements}$ = 2054), that vaccines are unsafe or population control ($n_{Engagements}$ = 1167), that the virus is human made and a bioweapon ($n_{Engagements}$ = 732), and

that China intentionally spread the virus to hurt other countries ($n_{Engagements}$ = 153). The Pearson's Chi-squared test of independence suggested that these counts differed statistically significantly, $\chi^2(5)$ = 58280, $P < 0.001$, Cramer's V = 0.035, 95% CI = 0.035, 0.036.

### Associations between participant characteristics and overall support for conspiracy theories
Next, we aimed to determine if the assessed variables predicted the likelihood that a user's social media engagement supported the different conspiracy theories. For this, the multi-level data were clustered at the participant level to prevent that some very active users disproportionately influenced the results. In addition, participants' number of followers and numbers of accounts they followed were controlled for. All $P$-values were Holm-corrected to adjust for the large number of tests and all continuous predictors were standardized to facilitate effect size interpretations. We estimated separate Bernoulli generalized linear mixed models for each of the six conspiracy theories.

In the first model (see Table 1), the Twitter (currently X) engagements of older participants were 86% more likely to support the conspiracy theory that the virus and response to it are aimed at creating economic instability and benefiting large corporations. Moreover, a quadratic relationship was found with respect to political orientation, suggesting that the engagements of participants self-identifying near the two endpoints of the political spectrum, and particularly at the very left, exhibited a greater tendency to support the theory compared to the engagements of politically moderate individuals (see Fig. 2).

In the second model (see Table 2), the Twitter engagements of older participants were 103% more likely to support the conspiracy theory that the public is being intentionally misled about the true nature of the virus and prevention. Furthermore, the Twitter engagements of participants who scored higher on the scale assessing belief in false information demonstrated a 32% increased likelihood of supporting the theory, while the Twitter engagements of those scoring higher in denialism exhibited a 26% greater likelihood to endorse the theory. As with the previous theory, a curvilinear association with political orientation was observed. The Twitter engagements of participants self-identifying at the endpoints of the political spectrum, and particularly those on the very right, exhibited a greater tendency to support the theory (see Fig. 2).

In the third model (see Table 3), the Twitter engagements of older participants were 114% more likely to support the conspiracy theory that the virus is human-made and a bioweapon. In addition, a linear effect of political orientation was observed, showing that the Twitter

**Table 1 | Conspiracy Theory 1: Standardized multi-level regression analysis predicting the likelihood that Twitter (currently X) engagements supported the conspiracy belief that the virus and response to it are aimed at creating economic instability and benefiting large corporations**

| Variable | β | 95% CI Lower | 95% CI Upper | OR | 95% CI Lower | 95% CI Upper | $P_{Holm}$ |
|---|---|---|---|---|---|---|---|
| (Intercept) | −10.82 | −11.29 | −10.35 | 0.00 | 0.00 | 0.00 | <0.001 |
| Followers on Twitter (currently X) | −0.19 | −0.78 | 0.41 | 0.83 | 0.46 | 1.50 | 1.000 |
| Following on Twitter (currently X) | 0.21 | −0.44 | 0.86 | 1.23 | 0.64 | 2.37 | 1.000 |
| Age | 0.62 | 0.43 | 0.82 | 1.86 | 1.53 | 2.26 | <0.001 |
| Education | 0.23 | 0.06 | 0.40 | 1.26 | 1.06 | 1.50 | 0.070 |
| Gender[a]: Woman | 0.07 | −0.28 | 0.42 | 1.07 | 0.75 | 1.52 | 1.000 |
| Gender[a]: Other | 0.18 | −0.82 | 1.17 | 1.19 | 0.44 | 3.24 | 1.000 |
| Political Orientation[b] (linear) | −0.29 | −0.55 | −0.04 | 0.74 | 0.58 | 0.96 | 0.207 |
| Political Orientation[b] (quadratic) | 0.27 | 0.09 | 0.45 | 1.31 | 1.09 | 1.56 | 0.030 |
| Political Party[c]: Republican | −0.34 | −1.00 | 0.32 | 0.71 | 0.37 | 1.38 | 1.000 |
| Political Party[c]: Independent | 0.19 | −0.25 | 0.64 | 1.21 | 0.78 | 1.89 | 1.000 |
| Political Party[c]: Other | 0.25 | −0.45 | 0.96 | 1.29 | 0.63 | 2.62 | 1.000 |
| Belief in False Information | 0.21 | −0.01 | 0.42 | 1.23 | 0.99 | 1.52 | 0.471 |
| Disbelief in True Information | −0.13 | −0.32 | 0.06 | 0.88 | 0.73 | 1.06 | 1.000 |
| Conspiracy Mentality | 0.05 | −0.16 | 0.27 | 1.06 | 0.85 | 1.31 | 1.000 |
| Narcissism | −0.13 | −0.31 | 0.05 | 0.88 | 0.74 | 1.05 | 1.000 |
| Denialism | 0.12 | −0.10 | 0.35 | 1.13 | 0.90 | 1.42 | 1.000 |
| Need for Chaos | −0.09 | −0.29 | 0.11 | 0.92 | 0.75 | 1.12 | 1.000 |
| Belief in Information Reliability | 0.00 | −0.18 | 0.18 | 1.00 | 0.83 | 1.20 | 1.000 |
| Importance of Verifying Information | 0.05 | −0.15 | 0.24 | 1.05 | 0.86 | 1.27 | 1.000 |
| Perceived Ability to Recognize Misinformation | 0.13 | −0.04 | 0.31 | 1.14 | 0.96 | 1.36 | 1.000 |
| Residual degrees of freedom | 7580235 | | | | | | |
| Adjusted ICC | 0.56 | | | | | | |
| Unadjusted ICC | 0.51 | | | | | | |
| Conditional $R^2$ | 0.60 | | | | | | |
| Marginal $R^2$ | 0.09 | | | | | | |

A generalized linear mixed model was estimated.

β Standardized regression coefficient, *CI* Confidence Intervals, *OR* Odds ratios based on standardized scores, $P_{Holm}$ One-tailed *P*-test, Holm-corrected for multiple comparisons, *ICC* Intraclass correlation coefficients, $R^2$ explained variance.

[a]Reference group = men.

[b]Higher values mean a more right-leaning political orientation.

[c]Reference group = Democrats.

engagements of more right-leaning participants were 62% more likely to support the theory.

In the fourth model (see Table 4), the Twitter engagements of older participants were 136% more likely to support the conspiracy theory that governments and politicians are intentionally spreading false information. Moreover, the Twitter engagements of participants scoring higher on the perceived ability to recognize misinformation were 17% more likely, and the engagements of those scoring higher on belief in false information 25% more likely to support the theory. As for the first and second conspiracy theory, a quadratic association with political orientation was observed. Engagements of participants on both ends of the political spectrum were more likely to support the theory (see Fig. 2).

In the fifth model (see Table 5), which concerned the conspiracy theory that China intentionally spread the virus to hurt other countries —the theory with very few positives (see Fig. 1)—none of the variables reached statistical significance.

In the final sixth model (see Table 6), the Twitter engagements of older participants were 102% more likely to support the conspiracy theory that the vaccines are unsafe or a means of population control. In addition, the engagements of those scoring higher on the scale assessing the tendency to believe in false information were 48% more likely to support the theory.

## Discussion

Leveraging a unique dataset, we demonstrate that combining artificial intelligence analyses of millions of individual behavioral data points with self-reported surveys can effectively pinpoint risk factors for phenomena evident in actual online behaviors. Taking support for conspiracy theories as a case in point, our study reveals that some, but not all previously proposed factors are statistically significantly associated with such behavior. This finding underscores the importance and viability of expanding research beyond mere self-reporting by harnessing the potential of big data.

In terms of demographics, older participants tended to support conspiracy theories more. Notably, age was the most consistent among all factors observed. This observation suggests that research should develop interventions targeting older cohorts of the population, for instance by promoting digital literacy. However, no evidence was found for education generally being a resilience factor or that being male constitutes a risk factor, contrasting with previous work[43,44].

Individuals who politically self-identified as very left or very right demonstrated a statistically significantly elevated behavioral support for three out of six conspiracy theories. This outcome coherently aligns with findings from a substantial, cross-cultural, self-report study that identified elevated levels of conspiracy mentality among

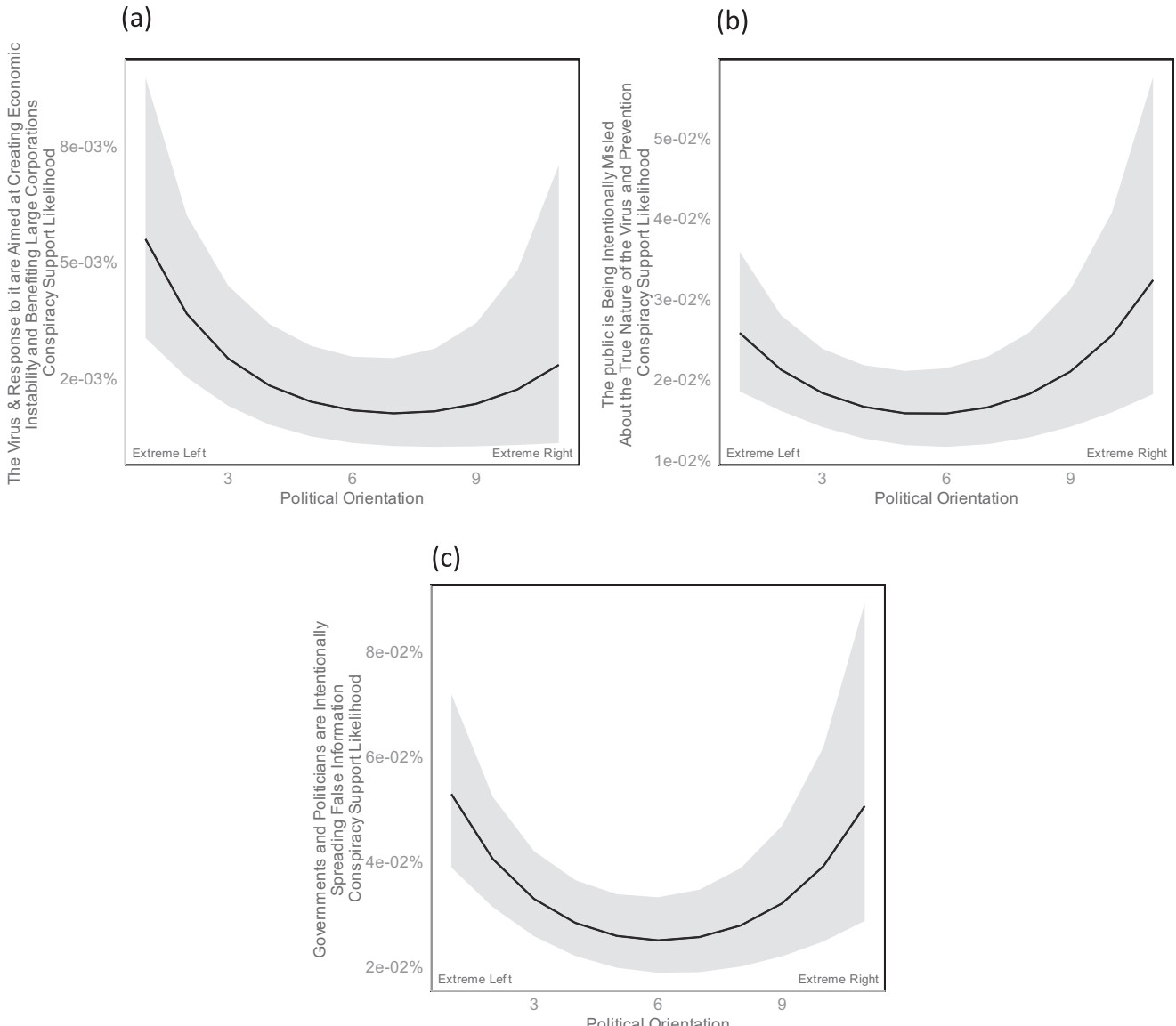

**Fig. 2 | Curvilinear associations between political orientation and specific conspiracy theory support likelihoods.** The y-axis shows the support likelihood for the given conspiracy theory at different levels of the x-axis, which represents participants' political orientation. Note that "extreme left" and "extreme right" reflect the endpoint response options of the scale. Panel (**a**) represents the support for the conspiracy theory that the coronavirus or the government's response to it are a deliberate strategy to create economic instability or to benefit large corporations over small businesses. Panel (**b**) represents the support for the conspiracy theory that the public is being intentionally misled about the true nature of the Coronavirus, its risks, or the efficacy of certain treatments or prevention methods. Panel (**c**) represents the support for the conspiracy theory that politicians or government agencies are intentionally spreading false information, or they have some other motive for the way they are responding to the coronavirus. The black line represents the estimated mean at a given x-axis level and ribbons represent 95% confidence intervals.

those situated at the endpoints of the political spectrum[22]. It also reinforces the results from extensive comparative studies, suggesting that neither side has a monopoly on conspiracy theorizing[25]. Notably, our study provides an expansion to existing research by positing that this phenomenon is also intrinsic to the dissemination of conspiracy theories on online social media platforms. Contradicting the view that particularly conservatives are chiefly responsible for the online proliferation of conspiracy theories[42,49–51], our findings imply that research should test interventions aiming to curb the spread of such theories among individuals on both ends of the political spectrum. However, even within these associations, nuances exist. The conspiracy theory suggesting that the virus and the response to it are part of an effort to benefit large corporations at the expense of small businesses found particularly strong resonance

among those at the far left of the political spectrum. Such theories, which emphasize economic exploitation, seem to align with the concern for equality by participants self-identifying politically at the very left. By contrast, the notion that the public is deliberately being deceived about the true nature of the virus and its prevention found the most support among those politically self-identified at the very right. Of note, for the theory positing that the virus is human-made and a bioweapon, political orientation displayed a linear association with the propensity to behaviorally endorse the theory, suggesting that individuals with right-leaning political views may be particularly susceptible. However, the conspiracy theory that governments and politicians are purposefully disseminating false information appealed comparably to both ends of the political spectrum[27].

**Table 2 | Conspiracy Theory 2: Standardized multi-level regression analysis predicting the likelihood that Twitter (currently X) engagements supported the conspiracy belief that the public is being intentionally misled about the true nature of the virus and prevention**

| Variable | β | 95% CI Lower | 95% CI Upper | OR | 95% CI Lower | 95% CI Upper | $P_{Holm}$ |
|---|---|---|---|---|---|---|---|
| (Intercept) | −8.74 | −9.03 | −8.45 | 0.00 | 0.00 | 0.00 | <0.001 |
| Followers on Twitter (currently X) | 0.01 | −0.44 | 0.47 | 1.01 | 0.64 | 1.60 | 1.000 |
| Following on Twitter (currently X) | −0.13 | −0.60 | 0.33 | 0.87 | 0.55 | 1.39 | 1.000 |
| Age | 0.71 | 0.58 | 0.84 | 2.03 | 1.78 | 2.31 | <0.001 |
| Education | 0.10 | −0.01 | 0.22 | 1.11 | 0.99 | 1.25 | 0.489 |
| Gender[a]: Woman | −0.05 | −0.28 | 0.19 | 0.96 | 0.76 | 1.20 | 1.000 |
| Gender[a]: Other | −0.17 | −0.94 | 0.60 | 0.84 | 0.39 | 1.82 | 1.000 |
| Political Orientation[b] (linear) | 0.01 | −0.16 | 0.18 | 1.01 | 0.85 | 1.20 | 1.000 |
| Political Orientation[b] (quadratic) | 0.21 | 0.09 | 0.33 | 1.23 | 1.10 | 1.39 | 0.003 |
| Political Party[c]: Republican | 0.06 | −0.34 | 0.45 | 1.06 | 0.71 | 1.57 | 1.000 |
| Political Party[c]: Independent | 0.28 | −0.02 | 0.58 | 1.32 | 0.98 | 1.79 | 0.487 |
| Political Party[c]: Other | −0.09 | −0.61 | 0.43 | 0.91 | 0.54 | 1.53 | 1.000 |
| Belief in False Information | 0.28 | 0.14 | 0.41 | 1.32 | 1.15 | 1.51 | 0.001 |
| Disbelief in True Information | −0.03 | −0.15 | 0.09 | 0.97 | 0.86 | 1.09 | 1.000 |
| Conspiracy Mentality | −0.02 | −0.16 | 0.12 | 0.98 | 0.85 | 1.13 | 1.000 |
| Narcissism | −0.11 | −0.23 | 0.01 | 0.90 | 0.80 | 1.01 | 0.489 |
| Denialism | 0.23 | 0.07 | 0.38 | 1.26 | 1.08 | 1.47 | 0.032 |
| Need for Chaos | −0.14 | −0.27 | −0.02 | 0.87 | 0.76 | 0.98 | 0.209 |
| Belief in Information Reliability | −0.03 | −0.15 | 0.09 | 0.97 | 0.86 | 1.10 | 1.000 |
| Importance of Verifying Information | 0.06 | −0.07 | 0.19 | 1.06 | 0.94 | 1.21 | 1.000 |
| Perceived Ability to Recognize Misinformation | 0.14 | 0.02 | 0.26 | 1.15 | 1.02 | 1.29 | 0.156 |
| Residual degrees of freedom | 7580235 | | | | | | |
| Adjusted ICC | 0.51 | | | | | | |
| Unadjusted ICC | 0.45 | | | | | | |
| Conditional $R^2$ | 0.56 | | | | | | |
| Marginal $R^2$ | 0.11 | | | | | | |

A generalized linear mixed model was estimated.

β Standardized regression coefficient, *CI* Confidence Intervals, *OR* Odds ratios based on standardized scores. $P_{Holm}$ One-tailed *P*-test, Holm-corrected for multiple comparisons, *ICC* Intraclass correlation coefficients, $R^2$ explained variance.

[a]Reference group = men.

[b]Higher values mean a more right-leaning political orientation.

[c]Reference group = Democrats.

In terms of psychological variables, the inclination to believe in false information[37] emerged as the most consistent factor, demonstrating a statistically significant and positive relationship with support for three conspiracy theories. This outcome is noteworthy as it endows the scale with predictive validity, offering researchers a concise and politically-balanced tool for evaluating tangible support for conspiracy theories. In the current study, the short version of the scale, which demonstrated only acceptable reliability, was employed. It is conceivable that the complete version of the scale could offer even greater predictive accuracy. Conversely, the tendency to reject true information did not significantly correlate with support for conspiracy theories. However, it is noteworthy that this scale showed low internal reliability (refer to Methods).

Denialism was associated with statistically significantly elevated support for the conspiracy theory that the public is being misled about the true nature of the virus and prevention. This finding provides some ecological validity to self-report findings implicating denialism in the belief in conspiracy theories[19]. It highlights that research should test interventions increasing people's reliance on and trust in official accounts of events to reduce the endorsement and spread of conspiracy theories online, especially when these theories directly imply that official public accounts are compromised.

Next, the perceived ability to discern misinformation on social media derived from the theory of reasoned action[29,30] was associated with statistically significantly elevated levels of support for the conspiracy theory that governments and politicians are intentionally spreading false information. The two additional theory of reasoned action factors, namely perceived importance of verifying social media information and the belief in the reliability of social media information, showed no statistically significant associations. Recognizing that confidence in one's ability to discern information's veracity is a key resilience factor in psychological inoculation frameworks[34,52,53], we conducted further tests to assess the robustness of our finding. Initially, we explored if the association between perceived ability and conspiracy theory support varied with participants' actual misinformation identification skills (i.e., scores on the scale assessing belief in false information; see Supplementary Information, Supplementary Note 1). However, no statistically significant interaction was found in this regard for any model. We also examined whether overconfidence as reflected in a curvilinear relationship might drive the observed association, but a statistically significant relationship did not emerge (see Supplementary Information, Supplementary Note 2).

It appears likely that confidence in detecting misinformation, as immediately evaluated in a task-specific inoculation intervention, is distinct from a broader perceived ability to identify misinformation. In

**Table 3 | Conspiracy Theory 3: Standardized multi-level regression analysis predicting the likelihood that Twitter (currently X) engagements supported the conspiracy belief that the virus is human-made and a bioweapon**

| Variable | β | 95% CI Lower | 95% CI Upper | OR | 95% CI Lower | 95% CI Upper | $P_{Holm}$ |
|---|---|---|---|---|---|---|---|
| (Intercept) | −11.84 | −12.48 | −11.20 | 0.00 | 0.00 | 0.00 | <0.001 |
| Followers on Twitter (currently X) | −0.26 | −1.08 | 0.56 | 0.77 | 0.34 | 1.75 | 1.000 |
| Following on Twitter (currently X) | 0.12 | −0.67 | 0.91 | 1.13 | 0.51 | 2.49 | 1.000 |
| Age | 0.76 | 0.52 | 1.00 | 2.14 | 1.68 | 2.73 | <0.001 |
| Education | 0.11 | −0.11 | 0.32 | 1.11 | 0.90 | 1.38 | 1.000 |
| Gender[a]: Woman | −0.45 | −0.88 | −0.02 | 0.64 | 0.42 | 0.98 | 0.280 |
| Gender[a]: Other | −0.39 | −2.04 | 1.27 | 0.68 | 0.13 | 3.55 | 1.000 |
| Political Orientation[b] (linear) | 0.48 | 0.17 | 0.80 | 1.62 | 1.18 | 2.23 | 0.027 |
| Political Orientation[b] (quadratic) | 0.07 | −0.14 | 0.28 | 1.07 | 0.87 | 1.32 | 1.000 |
| Political Party[c]: Republican | 0.35 | −0.40 | 1.10 | 1.42 | 0.67 | 3.00 | 1.000 |
| Political Party[c]: Independent | 0.82 | 0.22 | 1.42 | 2.26 | 1.24 | 4.12 | 0.068 |
| Political Party[c]: Other | 0.39 | −0.58 | 1.37 | 1.48 | 0.56 | 3.93 | 1.000 |
| Belief in False Information | 0.32 | 0.08 | 0.56 | 1.38 | 1.08 | 1.76 | 0.070 |
| Disbelief in True Information | 0.12 | −0.10 | 0.34 | 1.13 | 0.90 | 1.41 | 1.000 |
| Conspiracy Mentality | 0.06 | −0.21 | 0.34 | 1.07 | 0.81 | 1.40 | 1.000 |
| Narcissism | −0.25 | −0.47 | −0.03 | 0.78 | 0.62 | 0.97 | 0.200 |
| Denialism | 0.39 | 0.10 | 0.68 | 1.48 | 1.11 | 1.97 | 0.068 |
| Need for Chaos | −0.23 | −0.48 | 0.02 | 0.80 | 0.62 | 1.02 | 0.462 |
| Belief in Information Reliability | −0.09 | −0.31 | 0.13 | 0.92 | 0.74 | 1.14 | 1.000 |
| Importance of Verifying Information | −0.05 | −0.28 | 0.18 | 0.95 | 0.75 | 1.19 | 1.000 |
| Perceived Ability to Recognize Misinformation | 0.22 | 0.00 | 0.43 | 1.24 | 1.00 | 1.54 | 0.337 |
| Residual degrees of freedom | 7580235 | | | | | | |
| Adjusted ICC | 0.56 | | | | | | |
| Unadjusted ICC | 0.44 | | | | | | |
| Conditional $R^2$ | 0.66 | | | | | | |
| Marginal $R^2$ | 0.22 | | | | | | |

A generalized linear mixed model was estimated.

β Standardized regression coefficient, *CI* Confidence Intervals, *OR* Odds ratios based on standardized scores, $P_{Holm}$ One-tailed *P*-test, Holm-corrected for multiple comparisons, *ICC* Intraclass correlation coefficients, $R^2$ explained variance.

[a]Reference group = men.

[b]Higher values mean a more right-leaning political orientation.

[c]Reference group = Democrats.

an inoculation experiment, participants receive explicit instructions encouraging them to accurately judge the veracity of various types of information. By contrast, this study, which examined behavioral support for conspiracy theories on social media over several years, lacked such direct guidance as it observed naturally-occurring behavior. Furthermore, despite the absence of a statistically significant curvilinear relationship, it is possible that our measure of perceived abilities may still mirror participants' level of overconfidence, consistent with the Dunning-Kruger effect[54]. Building upon prior research and the findings of this study, future investigations should explore whether confidence developed through inoculation interventions over time feeds into or remains distinct from the broader self-perception of abilities evaluated here. This exploration is crucial for helping researchers identify which aspects of digital literacy programs might unintentionally produce adverse effects, like instilling a misguided sense of expertise in detecting misinformation, versus those that cultivate genuinely beneficial confidence.

The absence of a statistically significant finding with the conspiracy mentality scale[35] is somewhat perplexing. Unlike other scales that directly measure beliefs in a set of conspiracy theories, this tool gauges a general propensity towards such beliefs. One of the strengths of this approach is its avoidance of direct references to terms like "conspiracy theory." Such terminology is often objected to by actual believers in conspiracy theories and even perceived as part of a

conspiracy itself[55], potentially disrupting accurate measurement. Nevertheless, the scale's predictive validity has previously been called into question[23], partially due to its items being tangentially related to the core content of actual conspiracy theories or for tapping onto other constructs. For example, certain items (e.g., "Politicians usually do not tell us the true motives for their decisions." or "I think that government agencies closely monitor all citizens.") have been criticized as they might be considered factual in many societies or measure political trust rather than conspiracy mentality[56], possibly explaining the scale's frequent normal distribution. Future research, therefore, should explore whether scales that assess beliefs in specific theories might generate divergent results.

In our research, participants were asked to allow access to their Twitter (currently X) activity, which might suggest they have a lesser inclination towards conspiracy theories, given their willingness to trust researchers. Contrasting with such a view, the average conspiracy mentality score was notably above the midpoint on the normally-distributed scale (M = 7.04, SD = 1.87, Skewness = −0.10). Moreover, this mean falls within the general range seen across various studies[22] and is slightly higher than the mean in another quota-representative U.S. study conducted in the same period as the present study[57]. Furthermore, it is reasonable to assume that a potential selection bias would have similarly affected the variable of belief in false information, which showed statistically significant associations with three

**Table 4 | Conspiracy Theory 4: Standardized multi-level regression analysis predicting the likelihood that Twitter (currently X) engagements supported the conspiracy belief that governments and politicians are intentionally spreading false information**

| Variable | β | 95% CI Lower | Upper | OR | 95% CI Lower | Upper | $P_{Holm}$ |
|---|---|---|---|---|---|---|---|
| (Intercept) | −8.26 | −8.53 | −7.99 | 0.00 | 0.00 | 0.00 | <0.001 |
| Followers on Twitter (currently X) | −0.04 | −0.46 | 0.38 | 0.96 | 0.63 | 1.47 | 1.000 |
| Following on Twitter (currently X) | −0.04 | −0.50 | 0.42 | 0.96 | 0.61 | 1.52 | 1.000 |
| Age | 0.86 | 0.73 | 0.99 | 2.36 | 2.08 | 2.68 | <0.001 |
| Education | 0.12 | 0.01 | 0.23 | 1.13 | 1.01 | 1.26 | 0.217 |
| Gender[a]: Woman | 0.08 | −0.14 | 0.31 | 1.08 | 0.87 | 1.36 | 1.000 |
| Gender[a]: Other | 0.26 | −0.44 | 0.96 | 1.30 | 0.65 | 2.62 | 1.000 |
| Political Orientation[b] (linear) | −0.14 | −0.31 | 0.02 | 0.87 | 0.74 | 1.02 | 0.563 |
| Political Orientation[b] (quadratic) | 0.24 | 0.13 | 0.35 | 1.27 | 1.13 | 1.43 | <0.001 |
| Political Party[c]: Republican | −0.52 | −0.92 | −0.12 | 0.60 | 0.40 | 0.89 | 0.083 |
| Political Party[c]: Independent | −0.10 | −0.39 | 0.19 | 0.91 | 0.68 | 1.21 | 1.000 |
| Political Party[c]: Other | −0.34 | −0.84 | 0.15 | 0.71 | 0.43 | 1.17 | 0.962 |
| Belief in False Information | 0.22 | 0.09 | 0.35 | 1.25 | 1.09 | 1.42 | 0.012 |
| Disbelief in True Information | −0.07 | −0.18 | 0.05 | 0.94 | 0.83 | 1.05 | 1.000 |
| Conspiracy Mentality | 0.00 | −0.14 | 0.14 | 1.00 | 0.87 | 1.15 | 1.000 |
| Narcissism | −0.13 | −0.25 | −0.02 | 0.88 | 0.78 | 0.98 | 0.189 |
| Denialism | 0.03 | −0.12 | 0.18 | 1.03 | 0.89 | 1.19 | 1.000 |
| Need for Chaos | −0.07 | −0.19 | 0.06 | 0.93 | 0.82 | 1.06 | 1.000 |
| Belief in Information Reliability | −0.08 | −0.19 | 0.04 | 0.93 | 0.83 | 1.04 | 1.000 |
| Importance of Verifying Information | 0.09 | −0.03 | 0.21 | 1.09 | 0.97 | 1.24 | 0.931 |
| Perceived Ability to Recognize Misinformation | 0.16 | 0.05 | 0.27 | 1.17 | 1.05 | 1.31 | 0.040 |
| Residual degrees of freedom | 7580235 | | | | | | |
| Adjusted ICC | 0.51 | | | | | | |
| Unadjusted ICC | 0.44 | | | | | | |
| Conditional $R^2$ | 0.57 | | | | | | |
| Marginal $R^2$ | 0.13 | | | | | | |

A generalized linear mixed model was estimated.

β Standardized regression coefficient, *CI* Confidence Intervals, *OR* Odds ratios based on standardized scores, $P_{Holm}$ One-tailed *P*-test, Holm-corrected for multiple comparisons, *ICC* Intraclass correlation coefficients, $R^2$ explained variance.

[a]Reference group = men.

[b]Higher values mean a more right-leaning political orientation.

[c]Reference group = Democrats.

conspiracy theories. Therefore, selection bias does not seem to be a likely explanation for the statistically nonsignificant findings for conspiracy mentality in our study. It is further noteworthy that the discrepancy observed in our study between the normal distribution of the scale and the low number of Twitter (currently X) engagements supporting specific conspiracy theories aligns with common observations in this field of research[36]. Indeed, this divergence forms the foundation for conceptual critiques of what the conspiracy mentality questionnaire measures[58].

It is similarly noteworthy that both narcissism and need for chaos, previously identified as risk factors[14,17,20], did not emerge as statistically significant risk factors in this study. Owing to the intrinsic constraints associated with interpreting null findings within the context of inferential statistics, we refrain from drawing conclusions from these results. However, we strongly encourage subsequent research endeavors to further validate these measures with behavioral outcomes.

The current study holds broader implications for misinformation and conspiracy theory research. A central discussion in the field revolves around the question of whether predictors of susceptibility to misinformation and conspiracy beliefs identified in survey research are indicative of behavior in real-world settings[16]. Our findings support the role of some but not all the proposed variables by demonstrating that they are associated with pertinent online behaviors. Thus, our research

suggests that whereas studies based solely on self-report are not inherently devoid of ecological validity, a robust integration of self-report with machine learning analyses of extensive behavioral data might be crucial to affirm this validity.

It remains important to recognize that the overall variance explained in our sample was relatively low, ranging from 7% to 22%. This observation underscores the idea that factors other than those measured in this research may play a role. Whereas we attempted to include a broad range of the factors previously proposed to predict conspiracy beliefs, it was not feasible to include every possible variable. In future research, it may be beneficial to test the variables recently proposed in a meta-analysis on conspiracy beliefs[14], which was published after the participant-level data were collected for this study. In a similar vein, a recent study conducted in the United States, which examined a broad range of factors, suggested that variables such as populism, Manicheanism, and alignment with specific political figures (for example, Donald Trump) might be important[28].

Moreover, the intraclass correlation coefficient suggested that only about half the variance was overall attributable to the participant level for most of the theories (except for Conspiracy Theory 5 that concerned the actions of China, which had very few cases; we discuss reasons for this later). This reinforces the notion that large influences might stem from the characteristics of online social platforms rather

**Table 5 | Conspiracy Theory 5: Standardized multi-level regression analysis predicting the likelihood that Twitter (currently X) engagements supported the conspiracy belief that China intentionally spread the virus to hurt other countries**

| Variable | β | 95% CI Lower | 95% CI Upper | OR | 95% CI Lower | 95% CI Upper | $P_{Holm}$ |
|---|---|---|---|---|---|---|---|
| (Intercept) | −15.14 | −16.80 | −13.47 | 0.00 | 0.00 | 0.00 | <0.001 |
| Followers on Twitter (currently X) | −0.25 | −1.99 | 1.49 | 0.78 | 0.14 | 4.45 | 1.000 |
| Following on Twitter (currently X) | 0.20 | −1.49 | 1.89 | 1.23 | 0.23 | 6.65 | 1.000 |
| Age | 0.68 | 0.10 | 1.25 | 1.96 | 1.10 | 3.50 | 0.217 |
| Education | 0.12 | −0.38 | 0.63 | 1.13 | 0.69 | 1.87 | 1.000 |
| Gender[a]: Woman | −0.20 | −1.20 | 0.81 | 0.82 | 0.30 | 2.24 | 1.000 |
| Gender[a]: Other | −0.83 | −5.19 | 3.53 | 0.44 | 0.01 | 34.06 | 1.000 |
| Political Orientation[b] (linear) | 0.20 | −0.50 | 0.91 | 1.23 | 0.61 | 2.47 | 1.000 |
| Political Orientation[b] (quadratic) | 0.08 | −0.39 | 0.55 | 1.08 | 0.68 | 1.73 | 1.000 |
| Political Party[c]: Republican | −0.07 | −1.79 | 1.66 | 0.94 | 0.17 | 5.23 | 1.000 |
| Political Party[c]: Independent | 0.31 | −1.04 | 1.66 | 1.36 | 0.35 | 5.23 | 1.000 |
| Political Party[c]: Other | 0.15 | −2.08 | 2.39 | 1.16 | 0.12 | 10.87 | 1.000 |
| Belief in False Information | 0.50 | −0.06 | 1.05 | 1.64 | 0.94 | 2.86 | 0.774 |
| Disbelief in True Information | 0.26 | −0.25 | 0.78 | 1.30 | 0.78 | 2.17 | 1.000 |
| Conspiracy Mentality | 0.18 | −0.46 | 0.81 | 1.19 | 0.63 | 2.24 | 1.000 |
| Narcissism | −0.30 | −0.82 | 0.23 | 0.74 | 0.44 | 1.25 | 1.000 |
| Denialism | 0.20 | −0.44 | 0.85 | 1.23 | 0.64 | 2.34 | 1.000 |
| Need for Chaos | −0.21 | −0.78 | 0.36 | 0.81 | 0.46 | 1.43 | 1.000 |
| Belief in Information Reliability | 0.21 | −0.29 | 0.72 | 1.24 | 0.75 | 2.05 | 1.000 |
| Importance of Verifying Information | 0.02 | −0.52 | 0.55 | 1.02 | 0.60 | 1.74 | 1.000 |
| Perceived Ability to Recognize Misinformation | 0.13 | −0.37 | 0.64 | 1.14 | 0.69 | 1.90 | 1.000 |
| Residual degrees of freedom | 7580235 | | | | | | |
| Adjusted ICC | 0.85 | | | | | | |
| Unadjusted ICC | 0.79 | | | | | | |
| Conditional $R^2$ | 0.86 | | | | | | |
| Marginal $R^2$ | 0.07 | | | | | | |

A generalized linear mixed model was estimated.

β Standardized regression coefficient, *CI* Confidence Intervals, *OR* Odds ratios based on standardized scores, $P_{Holm}$ One-tailed *P*-test, Holm-corrected for multiple comparisons, *ICC* Intraclass correlation coefficients, $R^2$ explained variance.

[a]Reference group = men.

[b]Higher values mean a more right-leaning political orientation.

[c]Reference group = Democrats.

than traits of their users[59]. Previous studies indicate that structural features of social media platforms such as feedback cues (e.g., accuracy nudges) significantly influence the proliferation of misinformation online[60]. Additionally, social media platforms often promote repeated exposure to conspiracy theories, potentially leading to an increased perception of their accuracy over time[61].

Further, the individuals that users choose to follow and interact with can also account for variations in conspiracy theory support, as the dissemination of such content often hinges on the relationship with and trust in the individual sharing the information[62]. For instance, a recent study provides an intriguing perspective, tracking a substantial number of Reddit users longitudinally[15]. Initial interactions with individual conspiracy theorists and subsequent active recruitment by these individuals often preceded and led to increased participation in conspiracy theory groups. Additionally, ostracization from groups engaged in more conventional topics acted as a driving force toward such engagement. While Reddit's distinct group structure is not directly comparable to that of the Twitter (currently X) platform, employing longitudinal network analyses that combine survey and behavioral data from social media could be instrumental in understanding how changes in social networks over time are influenced by individual characteristics and vice versa. This approach could be particularly effective in mapping the temporal trajectories

and evolution of individuals who exhibit the risk factors identified in our study.

In addition to the architecture of social media platforms, the origin of the information, beyond the identity of the individual disseminating it, is influential. For example, research indicates that individuals who subscribe to conspiracy theories demonstrate a higher propensity to trust information originating from alternative sources as opposed to mainstream outlets[63]. This implies that the users' connection to the primary source of information, such as a news website, exerts an impact that could beneficially broaden the methodological framework employed in the current study.

It is crucial to acknowledge that the majority of Twitter (currently X) engagements we analyzed occurred during a period when the platform actively combated and banned COVID-19 misinformation[64]. Consequently, we likely missed capturing all users and engagements that overtly supported such conspiracy theories. This context should be borne in mind when interpreting the relatively low levels of conspiracy theory support identified in our study, and especially for the theory concerning China's intentional spread of the virus to hurt other countries. The absence of statistically significant predictors in the fifth model may be attributed to a low number of positive outcomes for this

**Table 6 | Conspiracy Theory 6: Standardized multi-level regression analysis predicting the likelihood that Twitter (currently X) engagements supported the conspiracy belief that the vaccines are unsafe or a means of population control**

| Variable | β | 95% CI Lower | 95% CI Upper | OR | 95% CI Lower | 95% CI Upper | $P_{Holm}$ |
|---|---|---|---|---|---|---|---|
| (Intercept) | −11.77 | −12.40 | −11.15 | 0.00 | 0.00 | 0.00 | <0.001 |
| Followers on Twitter (currently X) | 0.01 | −0.76 | 0.79 | 1.01 | 0.47 | 2.20 | 1.000 |
| Following on Twitter (currently X) | −0.21 | −0.98 | 0.56 | 0.81 | 0.38 | 1.76 | 1.000 |
| Age | 0.70 | 0.45 | 0.95 | 2.02 | 1.58 | 2.58 | <0.001 |
| Education | 0.03 | −0.18 | 0.25 | 1.03 | 0.83 | 1.28 | 1.000 |
| Gender[a]: Woman | −0.30 | −0.73 | 0.14 | 0.74 | 0.48 | 1.15 | 1.000 |
| Gender[a]: Other | −0.48 | −1.92 | 0.97 | 0.62 | 0.15 | 2.63 | 1.000 |
| Political Orientation[b] (linear) | 0.10 | −0.20 | 0.40 | 1.10 | 0.82 | 1.49 | 1.000 |
| Political Orientation[b] (quadratic) | 0.25 | 0.05 | 0.46 | 1.29 | 1.05 | 1.58 | 0.141 |
| Political Party[c]: Republican | −0.09 | −0.84 | 0.66 | 0.91 | 0.43 | 1.93 | 1.000 |
| Political Party[c]: Independent | 0.47 | −0.09 | 1.04 | 1.60 | 0.91 | 2.82 | 0.819 |
| Political Party[c]: Other | 0.44 | −0.45 | 1.33 | 1.56 | 0.64 | 3.79 | 1.000 |
| Belief in False Information | 0.39 | 0.15 | 0.64 | 1.48 | 1.16 | 1.89 | 0.016 |
| Disbelief in True Information | 0.12 | −0.11 | 0.34 | 1.12 | 0.90 | 1.41 | 1.000 |
| Conspiracy Mentality | 0.16 | −0.12 | 0.43 | 1.17 | 0.89 | 1.53 | 1.000 |
| Narcissism | −0.08 | −0.30 | 0.15 | 0.92 | 0.74 | 1.16 | 1.000 |
| Denialism | 0.32 | 0.03 | 0.61 | 1.37 | 1.03 | 1.83 | 0.257 |
| Need for Chaos | −0.07 | −0.29 | 0.16 | 0.94 | 0.75 | 1.17 | 1.000 |
| Belief in Information Reliability | 0.12 | −0.10 | 0.34 | 1.13 | 0.90 | 1.41 | 1.000 |
| Importance of Verifying Information | 0.12 | −0.12 | 0.36 | 1.13 | 0.89 | 1.44 | 1.000 |
| Perceived Ability to Recognize Misinformation | 0.16 | −0.06 | 0.38 | 1.17 | 0.94 | 1.46 | 1.000 |
| Residual degrees of freedom | 7580235 | | | | | | |
| Adjusted ICC | 0.62 | | | | | | |
| Unadjusted ICC | 0.54 | | | | | | |
| Conditional $R^2$ | 0.67 | | | | | | |
| Marginal $R^2$ | 0.12 | | | | | | |

A generalized linear mixed model was estimated.

β Standardized regression coefficient, *CI* Confidence Intervals, *OR* Odds ratios based on standardized scores, $P_{Holm}$ One-tailed *P*-test, Holm-corrected for multiple comparisons, *ICC* Intraclass correlation coefficients, $R^2$ explained variance.
[a]Reference group = men.
[b]Higher values mean a more right-leaning political orientation.
[c]Reference group = Democrats.

theory, a likely consequence of the platform's top-down filtering mechanisms at that time.

It is also vital to note that we focused on a smaller set of overarching conspiracy theories identified based on previous work[48]. While more detailed analyses involving an expanded set of conspiracy theories could offer nuanced insights, they might pose challenges in ensuring precise and distinct machine learning classifications. Future studies are also essential to ascertain whether the outcomes of our investigation can be replicated in the context of behavioral engagement with misinformation, as opposed to conspiracy theories, in the online realm. While an exhaustive examination of this possibility exceeds the scope of the current paper, which is explicitly centered on conspiracy theories, the observation that the scale assessing susceptibility to misinformation emerged as a relatively consistent factor implies that analogous mechanisms may be at work.

Although the sample showed close resemblance of many core demographic variables of the U.S. population on Twitter (currently X), it was not fully representative, somewhat limiting its generalizability. In particular, women, and people identifying as Democrats and Republicans were slightly overrepresented. In addition, our study focused on a specific social media platform. Variables with limited or inconsistent support based on our results might still be predictive on other platforms and in other contexts (e.g., offline).

Whereas our dataset addresses a significant gap in the literature by incorporating behavioral outcomes, it does not facilitate the discernment of causality. It is conceivable that the predictors under study influenced conspiracy theory support on social media; however, the inverse likely also holds true. For instance, individuals self-identifying politically as very left or very right may propagate and seek out conspiracy theories online that align with their views, further reinforcing their political polarization. This potential feedback mechanism warrants exploration in future research.

It is also crucial to acknowledge that the extent to which beliefs in conspiracy theories have significant consequences remains a contentious subject. A primary critique centers on the problematic extrapolation of causality from mere correlations[11]. While certain experimental and longitudinal studies indicate potential downstream impacts of conspiracy theories on social orientations and behavioral intentions[65,66], recent meta-analytic cross-lagged research has revealed only modest longitudinal effects, in this case in the context of health-related responses[1]. Our study is unable to investigate the downstream effects of endorsing conspiracy theories online in various life domains. However, the prevailing assumption of causality, predominantly derived from correlational studies, highlights this as a significant issue warranting further exploration[11].

Given that our hypotheses were directional in nature (e.g., predicting a positive association between conspiracy mentality and

## Table 7 | Participant Demographics

| | Study Sample | U.S. Twitter users[79] |
|---|---|---|
| Age M (SD) | 41.78 (14.87) | 39.90 (14.88) |
| **Gender in %** | | |
| Men | 45.8 | 50.35 |
| Women | 52.0 | 49.65 |
| Third gender/nonbinary/other | 2.10 | NA |
| **Employment in %** | | |
| Working full-time | 48.80 | 68.62 (combined) |
| Working part-time | 12.60 | |
| Unemployed and looking for work | 8.82 | 9.35 |
| A homemaker or stay-at-home parent | 7.50 | NA |
| Student | 5.75 | NA |
| Retired | 10.90 | 7.94 |
| Other | 5.47 | 7.31 |
| **Civil Status in %** | | |
| Married | 33.60 | 41.62 |
| Living with a partner | 12.40 | 8.07 |
| Widowed | 3.07 | 2.27 |
| Divorced/Separated | 10.60 | 9.60 |
| Never been married | 38.40 | 38.44 |
| **Education in %** | | |
| Less than high school degree | 1.36 | 3.72 |
| High school graduate (high school diploma or equivalent including GED) | 17.70 | 20.46 |
| Some college but no degree | 26.00 | 23.32 |
| Associate degree in college (2-year) | 12.90 | 10.19 |
| Bachelor's degree in college (4-year) | 28.90 | 26.35 |
| Master's degree | 10.50 | 12.52 |
| Doctoral degree | 1.00 | 3.43 (combined) |
| Professional degree (JD, MD) | 1.48 | |
| **Political Affiliation in %** | | |
| Republican | 24.80 | 20.88 |
| Democrat | 41.10 | 35.74 |
| Independent | 28.80 | 29.05 |
| Something else | 5.11 | 12.62 |
| **Race in %** | | |
| White | 77.60 | 77.58 |
| Black or African American | 12.40 | 11.32 |
| American Indian or Alaska Native | 1.36 | NA |
| Asian | 4.27 | NA |
| Native Hawaiian or Pacific Islander | 0.32 | NA |
| Other | 4.07 | 5.93 |

support for conspiracy theories), we opted for one-tailed tests. While one-tailed tests are sometimes perceived as less conservative compared to two-tailed tests, they provide a more powerful examination of directional predictions, while ensuring a logical alignment with hypotheses[67]. Furthermore, our statistical approach is already inherently conservative, incorporating P-value correction for a large number of tests. For instance, the P-value of perceived ability to detect misinformation in Table 4 increased from 0.002 to 0.040 after this correction.

In summary, using a unique dataset, our research underscores how combining artificial intelligence analyses of extensive behavioral data with self-reported surveys can help identify and validate proposed risk and protective factors for tangible online behaviors. With conspiracy theory support as our focal point, we discovered that only

select factors previously suggested may be statistically significantly associated with this behavior. This finding reinforces the critical need to move conspiracy theory and misinformation research beyond just self-reports, emphasizing the promise held by big data.

## Methods
The present research was approved by the Bioethics Committee of Jagiellonian University in Krakow (No 1072.6120.12.2022, January 26th, 2022). Informed consent was provided by all participants.

### Participants
Using the survey company CloudResearch, we recruited a sample of 2506 U.S. Americans that was intended to be representative. This sample size satisfied high statistical power to detect small level-2 effects in multi-level models according to guidelines[68], but no statistical method was used to predetermine sample size given the inherent computational challenges of multi-level simulations with millions of data points and complex data structures. Participant demographics are presented in Table 7. While the sample approached representativeness of the U.S. Twitter population in term of age, employment status, civil status, education, and race, participants identifying as women, and those identifying as Democrats, or as Republicans were slightly overrepresented. However, regarding gender representation (which was based on self-identification in our research), it is important to note that our study included a third gender option, which was not considered in the PEW report on the U.S. Twitter population. This additional category in our methodology could explain some of the observed differences in gender distribution when compared to the PEW findings.

### Procedure
The study drew upon data from two distinct sources. The initial phase, conducted between August 2022 and February 2023, involved participants completing an online survey questionnaire on Qualtrics. This survey encompassed various psychometric measures, including demographic questions reported in Table 7. Each participant began by signing an informed consent form, meticulously detailing the study's purpose, the data processing agreement that described participants' rights and data treatment aligned with GDPR, and the avenues to contact the researcher. Simultaneously, participants were asked about granting access to their Twitter (currently X) activities through an application custom-built for this research. By utilizing the app reliant on the Twitter (currently X) API, participants authorized us to monitor their current and historical engagement on the platform. They could withdraw this authorization at any time.

The application first scrutinized each Twitter (currently X) account's activity levels. Criteria for participation included a minimum of ten engagements in the past year (e.g., likes, posts, replies, reposts), having a public account, and maintaining an account at least two years old. This filtration process mitigated the inclusion of dormant accounts, while still reflecting the activity variability present within active Twitter (currently X) users.

Upon verifying the participants' accounts and obtaining access —a process completed within seconds—the participants proceeded with the specific psychometric instruments described in subsequent sections. Following the conclusion of this survey phase, we employed the academic Twitter (currently X) API to retrieve participants' activity during the COVID-19 pandemic, spanning from December 1, 2019, to December 31, 2021. Data extraction took place between January 10, 2023, and March 16, 2023, which was before the rebranding of Twitter to "X" (July 2023). Therefore, we refer to the platform as "Twitter (currently X)" throughout. We adhered to the terms of service of the platform when using the academic API. The resulting, comprehensive dataset comprised

7,713,506 Twitter (currently X) engagements (3,607,354 likes [any posts liked by the user]; 1,012,565 posts [content that the user personally posted, whether original or quoting another post; when quoting, the text written by the participants was retrieved, not the post commented on]; 1,084,863 replies [the content of any responses the user provided to other posts]; 2,008,724 reposts [posts that the user shared via reposting]). We subjected the content of these engagements to an analysis employing machine learning classification, the details of which will be elaborated later in this section.

## Psychometric instruments

Unless stated otherwise, responses were rated on Likert scales ranging from 1 (strongly disagree) to 7 (strongly agree). All scales have previously been validated in the referenced papers. The correlations between the main variables are visualized in Figure S2.

## Political orientation

On a Scale from 1 (extremely liberal/left-wing), through 6 (middle of the road) to 11 (extremely conservative/right-wing), participants indicated their political orientation.

## Political affiliation

Participants were asked to select their political affiliation from one of four options: "Democrat," "Republican," "Independent," or "Something else."

## Need for chaos

We used the 8-item need for chaos scale[20] (e.g., "I need chaos around me–it is too boring if nothing is going on."; $\alpha(2505) = 0.89$, 95% CI = 0.88, 0.90), capturing the psychological state in which individuals desire to create or promote chaos, particularly in the political domain.

## Narcissism

Narcissism was assessed with a four-item scale[69] (e.g., "I tend to want others to admire me."; $\alpha(2505) = 0.88$, 95% CI = 0.87, 0.89).

## Denialism

Denialism was assessed with a four-item scale[19] (e.g., "I often disagree with conventional views about the world."; $\alpha(2505) = 0.81$, 95% CI = 0.80, 0.83).

## Conspiracy mentality

We used the 5-item conspiracy mentality questionnaire[35] to index participants' general tendency toward conspiratorial thinking (e.g., "There are secret organizations that greatly influence political decisions."; $\alpha(2505) = 0.84$, 95% CI = 0.82, 0.85).

## Misinformation susceptibility

The 8-item version of the misinformation susceptibility scale[37] was used. Participants read the following introduction to the scale:

"Please categorize the following news headlines as either 'Fake News' or 'Real News.' Some items may look credible or obviously false at first sight but may actually fall in the opposite category. However, for each news headline, only one category is correct."

Next, participants were presented with eight statements, half of which were true (e.g., "Republicans Divided in Views of Trump's Conduct, Democrats Are Broadly Critical") or false (e.g., "Certain Vaccines Are Loaded with Dangerous Chemicals and Toxins"). For each they were asked to dichotomously rate whether they believed the statement was "fake" (1) or "real" (2). We separately calculated the tendency to perceive false information as real ($\alpha(2505) = 0.61$, 95% CI = 0.58, 0.63) and true information as false ($\alpha(2505) = 0.47$, 95% CI = 0.43, 0.50).

## Theory of reasoned action

Adopting a recent application of the Theory of Reasoned Action to misinformation susceptibility[30], we assessed participants' belief in the reliability of information (three items, e.g., "I trust information from my social networks; thus I do not have to check it."; $\alpha(2505) = 0.82$, 95% CI = 0.81, 0.83), attitudes toward verifying information (five items, e.g., "It is important to check the original source of information."; $\alpha(2505) = 0.86$, 95% CI = 0.85, 0.87), and perceived self-efficacy in recognizing misinformation (five items, e.g., "I can easily detect false information when I read it on social media."; $\alpha(2505) = 0.78$, 95% CI = 0.77, 0.80).

## Classification of engagements on the Twitter (currently X) platform

We employed a machine-learning approach to assess the behavioral support for COVID-19 conspiracy theories from Twitter (currently X) engagements. Our methodology encompassed several distinct steps:

1. **Selection of theories:** Drawing from previous research[48], we identified six overarching conspiracy theories that encapsulated the predominant conspiracies associated with COVID-19 (see Table 8).

2. **Identification of relevant engagements:** In our pursuit of developing a machine-learning framework adept at discerning and classifying the complex and elusive characteristics of conspiracy theory endorsement, we implemented a semantic similarity search to pinpoint interactions topically related to these theories. This strategy was pivotal from a resource management perspective, as it facilitated a significant reduction in the computational power required for assessing conspiracy theory support in the subsequent phase. The second model, delineated in the ensuing numbered section, demands considerably more resources, as we relied on a commercial-grade model. Estimating the support for conspiracy theories across all 7.7 million Twitter (currently X) engagements would have substantially surpassed our financial limits. Therefore, this method was essential in enabling the execution of the second, more demanding model within the financial boundaries of our research project. As we subsequently illustrate through validation processes, the selected similarity threshold ensured with considerable certainty that the preponderance of Twitter (currently X) engagements endorsing conspiracy theories in the overall dataset was accurately identified.

   Specifically, we utilized a sentence transformer model (sentence-transformers/all-mpnet-base-v2)[70] to compute embeddings for each engagement and conspiracy theory, which maps a given text to a 768-dimensional dense vector space. The similarity score is calculated from the distance (squared Euclidean distance) between the engagement vector and the conspiracy theory vector; it ranges from 0 (no resemblance) to 1 (identical). For demonstration, consider the following two posts with relatively high semantic similarity scores regarding the conspiracy theory that vaccines are unsafe and an effort to control and reduce the population (our first theory, see Table 8). The post, "Bill Gates has been talking about population control openly for years–now we have a coronavirus vaccine! Coincidence? I think not!" has an estimated semantic similarity of 0.615, and the post, "Multiple variants of COVID-19 have emerged. Vaccines are our shield. Don't let conspiracies disarm you. #FightTheVirus," an estimated semantic similarity of 0.664. Thus, both posts are correctly identified as topically related to the conspiracy theory. By contrast, the post, "Sunday (tomorrow) is National Ice Cream Day and have we got a gift for you! Join us for an ice cream sundae," has an estimated semantic similarity score of 0.031, rending it entirely unrelated. To facilitate efficient search of

**Table 8 | Model Validation for Both the Similarity and Support Classification Models, and Performance Estimates for the Support Classification Model**

| | Similarity Classification Model | | Support Classification Model | | | | |
|---|---|---|---|---|---|---|---|
| | Human Interrater Reliability α | r Similarity Model Prediction – Average Human Annotation | Mean PABAK: Human Support Annotation | PABAK: GPT – Human Majority Vote | Precision | Recall | F1 |
| 1. The coronavirus or the government's response to it is a deliberate strategy to create economic instability or to benefit large corporations over small businesses. | 0.90 | 0.59 | 0.83 | 0.93 | 0.83 | 0.67 | 0.74 |
| 2. The public is being intentionally misled about the true nature of the Coronavirus, its risks, or the efficacy of certain treatments or prevention methods. | 0.90 | 0.71 | 0.66 | 0.75 | 0.74 | 0.63 | 0.68 |
| 3. The coronavirus was created intentionally, made by humans, or as a bioweapon. | 0.95 | 0.42 | 0.89 | 0.87 | 0.82 | 0.61 | 0.70 |
| 4. Politicians or government agencies are intentionally spreading false information, or they have some other motive for the way they are responding to the coronavirus. | 0.93 | 0.65 | 0.76 | 0.79 | 0.59 | 0.85 | 0.70 |
| 5. The Chinese government intentionally created or spread the coronavirus to harm other countries. | 0.94 | 0.78 | 0.93 | 0.91 | 0.73 | 0.69 | 0.71 |
| 6. The coronavirus vaccine is either unsafe or part of a larger plot to control or reduce the population. | 0.90 | 0.77 | 0.89 | 0.94 | 0.80 | 0.73 | 0.76 |

α Cronbach's alpha, PABAK Prevalence-Adjusted Bias-Adjusted Kappa, r Pearson's correlation coefficient.

engagements topically related to the conspiracy theories, we built a FAISS index[71].

In response to the limitations of our resources, we established a similarity score threshold of 0.25. This decision was strategic and data driven, aiming to exclude completely irrelevant Twitter (currently X) engagements that would demand significantly higher computing resources, while still maintaining a high level of inclusivity. The threshold was carefully chosen to balance our resource constraints against the risk of Type 2 errors, which involve missing true positives. At this stage, Type 1 errors, which result in including false positives, were not a primary concern as we planned to conduct a detailed analysis of the conspiracy theory support among all Twitter (currently X) engagements scoring above 0.25 in a subsequent phase with a different model.

To enhance the validity of our methodology, we enlisted three raters to annotate a stratified random sample of 1,019 Twitter (currently X) engagements, selected from a corpus of millions of tweets gathered from participants' social networks. This sample was constituted by randomly allocating 50 Twitter (currently X) engagements for each conspiracy theory across four model similarity score ranges: 0.00–0.25, 0.25–0.50, 0.50–0.75, and 0.75–1.00. The resultant sample size being less than the anticipated 1200 (calculated from 50 engagements × 6 conspiracy theories × 4 similarity brackets) is attributed to the occurrence of fewer than 50 engagements for some theories within the highest similarity bracket of 0.75–1.00. This shortfall was anticipated, as achieving the highest similarity scores necessitates close to linguistic equivalence. The raters evaluated the topical relevance of these engagements in relation to their respective conspiracy theories, employing a four-point scale: 1 signifying "Topically unrelated," 2 "Likely topically unrelated," 3 "Likely topically related," and 4 "Topically related." The interrater reliability (calculated as Cronbach's alpha) was excellent across each conspiracy theory, and the mean ratings exhibited moderate to strong correlations with the model similarity estimates (see Table 8, first two columns), thereby affirming the model's validity. Crucially, the findings revealed that the model similarity threshold of 0.25 captured all ground truth positives based on the human annotator majority vote (both leniently defined as at least two ratings of 3 or higher, see upper panel of Figure S1, and conservatively defined as at least two ratings of 4, see lower panel of Fig. S1 in the Supplementary Information).

Using this cutoff, we identified the following relevant Twitter (currently X) engagements among the 7.7 million engagements with similarity scores above 0.25 for at least one conspiracy theory, which were subjected to the support classification model presented below: 462,452 engagements for Theory 1, 326,380 for Theory 2, 369,568 for Theory 3, 565,274 for Theory 4, 291,514 for Theory 5, and 381,835 for Theory 6. This proportion of the 7.7 million overall participant engagements on the platform during the COVID-19 pandemic might first appear substantial. However, the similarity cutoff was deliberately set to be inclusive, to minimize Type-2 errors in detection. As displayed in Fig. 1, the numbers of actual positives were much smaller. Thus, the vast majority of engagements with a similarity score above 0.25 do neither support (see Fig. 1) nor concern a given conspiracy theory (see Supplementary Information, Figure S1).

3. **Model prompt engineering and validation:** For the subsequent evaluation of the support classification model, we initially created a ground truth test dataset. Specifically, three evaluators annotated a series of engagements extracted from the Twitter (currently X) engagement within the social networks of

participants, rather than the participants themselves. We selected 200 engagements per conspiracy theory for this purpose. Notably, considering the prior validation of the similarity model, which demonstrated that a majority of positive instances were identified at higher levels of similarity (see Supplementary Information), we intentionally oversampled engagements exhibiting higher similarity scores. For each conspiracy theory, we included 25% of engagements with similarity scores ranging from 0.25 to 0.45, another 25% with scores from 0.45 to 0.65, and the remaining 50% with scores exceeding 0.65. This approach yielded a total of 1136 Twitter (currently X) engagements after the exclusion of tweets in other languages than English. The agreement between raters was substantial for two theories and excellent for four theories (see Table 8, third column).

Next, utilizing the most recent OpenAI GPT 3.5 model (gpt-3.5-turbo-0125, temperature = 0), we devised six prompts. Each prompt was designed to ascertain whether a given Twitter (currently X) engagement expressed support for one of six conspiracy theories. Initially, the prompts stated the respective conspiracy theory under consideration, followed by a comprehensive exposition delineating the criteria for what constituted endorsement or rejection of the theory (see Supplementary Online Materials for the full prompts). This delineation was grounded in the preparatory training and guidelines provided to coders prior to the manual annotation of the ground truth dataset. Subsequently, the model was instructed to furnish whether a tweet supported the respective theory in a binary format of "YES" (recoded as 1) and "NO" (recoded as 0).

Next, we undertook a multifaceted validation of the model prompts. Initially, we assessed the concordance between the GPT model's responses and the ground truth data. This phase of validation revealed excellent agreement between machine and human evaluations for four of the conspiracy theories, and substantial agreement for the remaining two (refer to Table 8, fourth column). Following this, we appraised the performance metrics of the model. The precision metrics were deemed satisfactory, and the recall rates were considered acceptable for all theories except one. The exception was the fourth conspiracy theory. However, whereas it exhibited a relatively lower precision metric, the recall was high, compensating for that imprecision. Crucially, across all theories, the F1 scores were either closely approaching or surpassing 0.7, indicative of decent model performance in a task as nuanced as discerning support for conspiracy theories within brief Twitter (currently X) engagements.

4. **Estimation of support:** In light of the model's satisfactory performance, we employed the prompts to categorize all instances of Twitter (currently X) engagements associated with a particular conspiracy theory, provided that the respective similarity score exceeded 0.25. To maintain the sample's representativeness by utilizing the full dataset, all instances of Twitter (currently X) engagements which exhibited a similarity score below 0.25, were assigned a support value of 0 (denoting "NO"). This adjustment was predicated on the validation outcomes of the similarity model, which indicated a significant improbability of true positives falling beneath this threshold. Nevertheless, for the purpose of transparency, the outcomes derived from a sample solely consisting of Twitter (currently X) engagements with similarity scores surpassing 0.25 are presented in the Supplementary Information.

5. **Integration with psychological data:** Finally, we merged the Twitter (currently X) engagement dataset with the individual-level psychological data previously detailed. This resulted in multi-level data structures, wherein Twitter (currently X) engagements (level 1) were nested within individual participants (level 2). The scores from the psychological measures attributed to the participants were therefore positioned at the latter second level. This integration allowed for a comprehensive analysis, juxtaposing Twitter (currently X) behavior with individual psychological profiles.

## Analyses

We used the glmmTMB[72] package v. 1.1.9 to estimate Bernoulli multi-level generalized linear models in R 4.2.2[73] to test for the effects of the psychological variables (level 2) on the Twitter (currently X) engagement estimates (level 1). The performance[74] package v. 0.10.1 was used to estimate intraclass correlation coefficients (ICC) and explained variance ($R^2$). Importantly, to enable the comparison of effect sizes, all predictors except for categorical factors (e.g., gender), were centered and standardized. In addition to the standardized coefficients, we report odds ratios for interpretability. Ggplot[75] v. 3.4.0 and Ggeffects[76] 1.1.5 were used for graphs. Correlations did not suggest muti-collinearity (see Fig. S2). All reported $P$-values in the main models are one-tailed as predictions were unidirectional and Holm-corrected to adjust for multiple tests. No data were excluded other than cases with missing values that were excluded listwise. The assumptions underlying the models were evaluated using version 0.4.6 of the DHARMA package[77], and the detailed outcomes are delineated in the Supplementary Information. The dispersion analysis yielded non-significant results for Models 1, 2, 4, 5, and 6, suggesting homoscedasticity. Conversely, for Models 3 and 5, the dispersion test was significant, with a dispersion value below 1, indicating underdispersion. This underdispersion implies that the statistical tests of significance applied to these models adopt a more conservative stance due to potentially larger standard errors.

### Reporting summary

Further information on research design is available in the Nature Portfolio Reporting Summary linked to this article.

## Data availability

The data[78] utilized in this study, required for replicating our statistical analyses, have been deposited in anonymized form in the Open Science Foundation database at https://doi.org/10.17605/OSF.IO/XPVFZ. The raw Twitter (currently X) data are available under restricted access in adherence to GDPR and the stipulations of the ethics approval and the data management board (J.R.K., J.P., R.R., M.M.), which include a Data Sharing Agreement that mandates secure storage and anonymization for open access, while prohibiting third-party sharing and requiring compliance with data privacy laws. To maintain transparency and reproducibility, we offer remote access to the data (safe haven data sharing model). Access can be obtained by contacting the corresponding author. We aim to process and respond to requests within a month.

## Code availability

The code required for replicating our results can be accessed at https://doi.org/10.17605/OSF.IO/XPVFZ.

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

## Acknowledgements

The research leading to these results has received funding from the EEA Financial Mechanism 2014-2021. Project registration number: 2019/35/J/HS6/03498. Artificial intelligence tools building on large language models were utilized to improve the writing of this manuscript. The authors take full responsibility for the content.

## Author contributions

Conceptualization: J.R.K., A.B.G., I.K., T.W., J.P., R.R., S.L., M.M. Methodology: J.R.K., I.K., A.B.G., T.W., J.P., R.R., M.M. Investigation: J.R.K., I.K. Visualization: J.R.K. Funding acquisition: J.R.K., J.P., R.R., M.M. Writing—original draft: J.R.K. Writing—review and editing: J.R.K., I.K., A.B.G., T.W., J.P., R.R., S.L., M.M. J.P. and T.W. contributed equally.

## Competing interests

The authors declare no competing interests.
