## [Peer Review File · Nature Communications]

Leveraging Artificial Intelligence to Identify the Psychological Factors Associated With Conspiracy Theory Beliefs OnlineREVIEWER COMMENTS

Reviewer #1 (Remarks to the Author):

In this research, the authors investigate the associations between psychological predispositions and support for conspiracy theories on the social media platform X. I was excited to review this paper. It uses a novel methodology to address a problem that is of interest to many social scientists (as well as the public): who engages with conspiracy theories. Explaining real-life social media conspiracy theory engagement is a major strength of the paper. Still, as typical, I had a few questions and comments about the manuscript.

1) I admit that, just like the authors, I was perplexed by the non-significant effects observed for conspiracy mentality. As the authors note, this measure has been occasionally criticized for capturing other predispositions, such as levels of political trust, but this does not mean we would expect it to be irrelevant for engagement with conspiracy theories. This almost seems like a sanity check on the dataset, and the fact that this effect was not observed makes me wonder about one major issue in particular: to what degree the study set up might have affected the psychological make up of the final sample.

Participants were asked to grant the researchers access to their X activities through a custom-built app. I would imagine this is more likely to be acceptable for participants who are overall trusting towards scientists or have low levels of conspiracy mentality. Thus, the current results might be more reflective of those who are generally less likely to engage with conspiracy theories. This is consistent with the current results showing that the average likelihood that X engagements supported conspiracy theories was generally low. It would be good to gauge somehow if these levels are comparable to those generally observed on X in the same period among other participants (e.g., those who had public profiles but did not take part in the study). It would also be useful to see if the distribution of variables such as conspiracy mentality in this sample is similar to that observed in other samples.

2) Furthermore, I appreciated that the model identifying conspiracy theories engagements has been refined, and that the new model "surpassed the untuned model's performance". However, the correlations with human evaluations for the fine-tuned model were still modest, especially for certain conspiracy theories.

3) I also admit I found integration of the theory of reasoned action a bit confusing. These variables are based on past work, but I think that the reasoning could be explained a bit better here. Authors predict (and find) that perceived self-efficacy in recognizing misinformation is associated with support for conspiracy theories. This is consistent with past theorising by Khan & Idris (2019) who linked such self-efficacy perceptions to sharing information without verification on social media. But couldn't one equally convincingly argue that perceived self-efficacy in recognizing misinformation could go hand in hand with a lower tendency to share it? For example, Basol et al. (2020; J Cogn) found that playing Bad News (the online fake news game in which players learn about common misinformation techniques) not only improved the ability to spot misinformation techniques but also increased people's level of confidence in their own judgments. Also, given work by Lyons et al. (2021) linking false news susceptibility to *over*confidence in news judgments, it might be important to consider self-efficacy in relation to ability. One idea could be to examine whether there is an interaction effect between the self-efficacy measure and the misinformation susceptibility indices (although I appreciate this is tricky given poor reliability of the latter measures). Overall, I think that the association between perceived self-efficacy in recognizing misinformation and support for conspiracy theories could use a deeper discussion, especially given that it was one of the few significant associations observed in the current study.

4) Furthermore, it was not clear to me if the effects that were observed in the study were unique to engagement with conspiracy theories or with misinformation more broadly. This might be a nuanced

but important distinction, particularly for predictors such as perceived self-efficacy in recognizing misinformation.

5) It would also be useful to account for participants' overall engagement with X, for instance by accounting for the number of followers.

6) As the authors note, the current methodology "does not facilitate the discernment of causality", so I would encourage them to be mindful of using causal language across the text (e.g., p. 5 reads: "We proceeded to evaluate the influence of various individual difference factors.").

7) P. 13 reads: "attributed to men's higher tendency to subjectively perceive themselves as a marginalized minority group." – this statement should be supported by a reference.

8) All p-values in the analyses are one-tailed "as predictions were unidirectional". I would have been more comfortable with this choice if the analytic strategy was pre-registered.

Reviewer #2 (Remarks to the Author):

I thank the authors for this important work on how psychological factors relate to the actual support of conspiracy theories on X. Given the prevalence of conspiracy theories online and the concerns about their potential adverse societal outcomes, this work addresses significant and momentous issues. In particular, this work combines surveys with behavioral data to provide empirical evidence of the risk and protective factors related to public endorsement of conspiracy theories. Leveraging a panel of over two thousand X users, subjected to assessment for several conspiracy support factors identified in the literature, the authors analyze how such factors are related to their support for six COVID-19-related conspiracy theories on X, as signified by their posts, replies, likes, and shares. In particular, the authors use machine learning techniques to estimate the dependent variable---the likelihood of supporting each of the six theories. Combining measures of individuals' psychological attitudes and beliefs with observational data in assessing actual support for conspiracy theories online is, to the best of my knowledge, a novel and important addition to the literature.

Overall, the manuscript is well-written and easy to follow. Although the analyses appear sound, and the results presented in the manuscript have the potential to inform future research and policy alike, certain methodological details prevent me from fully endorsing its publication in its current form.

Construction of the dependent variable

- If I understood correctly, the final dataset used for producing the results in tables 1 and 2 are the ~800k engagements that pass the filter described in the section "Identification of relevant engagements", rather than the full 12M-strong corpus. This leads me to a few considerations.
- First, I suggest that the authors report the number of users who engage with each of the six theory topics, as users may decide not to engage with any, or to discuss all of them. Preventing doubts about bias that may be introduced by subsampling engagements could further strengthen the manuscript.
- Second, the authors could further clarify how to best interpret the dependent variable, which is crucial to assessing the overall contribution. The fact that the sampling universe used to compare the variety of the measure is restricted, the measure itself may not capture the likelihood of a user expressing support for a theory, as most of their expressions are excluded from the analysis. Is the aim of the dependent variable to measure, when users engage with a topic related to a theory, the degree to which they support the theory? This point relates to my suggestions about further unpacking the results of the semantic similarity subtasks.

- Third, to better understand the results it would be useful to know which output of the deBERTa model was used. Given that the model was fine-tuned on a three-class dataset (entail, contradict, neutral), was it just the output related to entailment, or were the two other outcomes also taken into consideration? This potentially changes the meaning of the lower end of the measure (opposition to the theory, neutral stance towards it, irrelevance of the content with respect to the theory, etc.).
- Fourth, on a minor note, clarifying the choices of sampling universe may also further contextualize Figure 1. ~2--15% likelihood of support for each of the six theories is, in absolute terms, very large, and may lead to misunderstanding if quoted out of such context.

The role of different types of engagement

- Surprisingly, conspiracy mentality seems significantly *negatively* associated with support of conspiracy theories in posts, and similarly, belief in false information appears *negatively* associated with them in replies. I would argue that posts and replies are the most reliable signals, out of the four engagement types, of users' own beliefs in a theory or support for it. If I am reading the SI correctly, I wonder: does this entail that users with a higher conspiracy mentality and lower ability to detect misinformation are less likely to express public support for conspiracy theories on X --- but only if they are initiating a conversation? This is counterintuitive to me but potentially interesting in relation to the use and characteristics of the platform. To this end, relating engagement styles with psychological profiles may be useful to unpack the surprising results.
- More broadly, given the relative heterogeneity of the engagement types and that results may be dominated by more frequent, lower-effort interactions (likes and reposts, which make up three-quarters of the final dataset), I am curious about whether inter-subject differences may be completely mediated by engagement type, or if conversely, engagement type and individual profiles show structural associations.

Validation of the machine learning models

- The area where I feel the authors could significantly strengthen the readers' confidence in the presented results is explaining the integration of machine learning models with survey data. I recognize that this intersection is one of the methodological novelties of the manuscript, therefore I think it is important that it is conveyed appropriately. In particular, there is very little discussion about the validation of the outcomes of the machine learning models on two sophisticated tasks: determining whether a piece of content is relevant to the semantic area of a conspiracy theory, and the degree to which it expresses support to the theory.
- First, the description of the methodology would be clearer if the authors could unpack why the two tasks are necessary. For example, considering that the authors already make use of commercial generative AI for creating samples to finetune the latter model, could they not have used it directly to infer entailment between a contribution and a theory, perhaps to even more accurate results?
- Second, since the model for semantic similarity appears crucial in determining the dataset on which to infer support for conspiracy theories, it would be important to evaluate its output qualitatively (e.g., does it capture general topics like government intervention in COVID times, or more precisely the issues of government misconduct relevant to the theories?) and quantitatively (e.g., how accurate is it?). As a byproduct, evaluating this model would provide an important description of the dependent variable of the study.
- Third, there is little description of the validation of the model used to assess support for the theories. This is my main concern with the manuscript in its current form, as this model defines the main dependent variable of the study. The evaluation consists of correlation measures between automated and human labels on ~50--60 samples per theory (of which, a minority are expected to express full support for the theory). Humans The authors could clarify what correlation measure they use to estimate IRR between the five raters, and if they aggregated human judgments when compared with automated measures. Most importantly, especially in light of the low number of validation examples and the out-of-sample data used to finetune the model, it would be important to strengthen confidence in the accuracy of this model. Establishing baselines and testing the model on ground-truth data may be two ways in which the authors could address this concern. I am adding, for reference, some of the existing resources to this end:

Gambini, M., Tardelli, S. & Tesconi, M. The Anatomy of Conspirators: Unveiling Traits using a Comprehensive Twitter Dataset. arXiv (2023).

Langguth, J. et al. COCO: an annotated Twitter dataset of COVID-19 conspiracy theories. J. Comput. Soc. Sci. 1–42 (2023) doi:10.1007/s42001-023-00200-3.

Moffitt, J. D., King, C. & Carley, K. M. Hunting Conspiracy Theories During the COVID-19 Pandemic. Social Media and Society 7, (2021).

Phillips, S. C., Ng, L. H. X. & Carley, K. M. Hoaxes and Hidden agendas: A Twitter Conspiracy Theory Dataset. in WWW Companion (2022). doi:10.1145/3487553.3524665.

On a minor note, the authors state that studies that endeavored to identify determinants of engagement with conspiracy theories have rarely made use of large-scale behavioral insights. While I agree that the combination of survey measures and large-scale analyses in this field is rare, there are several studies addressing precursors to engagement with online conspiracy theories or support for them, such as:

Nera, K., Leveaux, S. & Klein, P. P. L. E. A "Conspiracy Theory" Conspiracy? A Mixed Methods Investigation of Laypeople's Rejection (and Acceptance) of a Controversial Label. Int. Rev. Soc. Psychol. 33, (2020).

Phadke, S., Samory, M. & Mitra, T. What Makes People Join Conspiracy Communities?: Role of Social Factors in Conspiracy Engagement. Proceedings of the ACM on Human-Computer Interaction CSCW, (2020).

Reviewer #3 (Remarks to the Author):

Here are my comments:

1. It isn't clear how causal conspiracy theories, or beliefs in conspiracy theories are. I am sure they are sometimes, but not all of the time. The authors should be more circumspect about their role in driving outcomes. See for example: Uscinski, Joseph, Adam M. Enders, Casey Klofstad, and Justin Stoler. 2022. "Cause and Effect: On the Antecedents and Consequences of Conspiracy Theory Beliefs." Current Opinion in Psychology 101364.

2. Page 3. Conspiracy theories are not always "intricate". The authors should settle on a published definition and cite it on the top of page 3.

3. Page 3. There is a rich literature on conspiracy theories and the factors associated with belief in them. You might want to remove the "misinformation studies" section on page 3 and focus on conspiracy theory studies. This current study seems to be about conspiracy theories, so it makes sense to motivate it from that rich literature. See for example:

Brandenstein, Nils. 2022. "Going Beyond Simplicity: Using Machine Learning to Predict Belief in Conspiracy Theories." European Journal of Social Psychology 52: 910-30.

Uscinski, Joseph, Adam Enders, Amanda Diekman, John Funchion, Casey Klofstad, Sandra Kuebler, Manohar Murthi, Kamal Premaratne, Michelle Seelig, Daniel Verdear, and Stefan Wuchty. 2022. "The Psychological and Political Correlates of Conspiracy Theory Beliefs." Scientific Reports 12: 21672.

It comes off odd that the authors ignore much of the conspiracy theory literature and then say that because the misinformation literature is not about conspiracy theories, there is a need for the present study (bottom of page 3). I think the authors should dump much of the misinformation literature and motivate this study from the ct literature since that is what this study is about.

4. This same issue comes up again on page 4 where the authors claim that previous research (cites 23 and 24) show that republicans engage more with conspiracy theories. But, as the authors already said, these studies are about untrustworthy websites and misinformation, not conspiracy theories specifically.

5. Some good studies of ct engagement to look at are:

Bessi, Alessandro, Mauro Coletto, George Alexandru Davidescu, Antonio Scala, Guido Caldarelli, and Walter Quattrociocchi. 2015. "Science Vs Conspiracy: Collective Narratives in the Age of Misinformation." PLoS ONE 10: e0118093.

Mancosu, Moreno, and Federico Vegetti. 2020. "'Is It the Message or the Messenger?': Conspiracy Endorsement and Media Sources." Social Science Computer Review.

Miani, Alessandro, Thomas Hills, and Adrian Bangerter. 2021. "Loco: The 88-Million-Word Language of Conspiracy Corpus." Behavior Research Methods.

6. The paper glosses over the data collection effort very quickly. It seems like the 2300 users averaged 5000 interactions with these six conspiracy theories. That seems like a lot. Could the authors explain this a bit? Are these 2272 people representative of the US? Can we generalize from this sample? Were they picked because they are already known to be conspiratorial?

7. The study focuses on a handful of traits that have been found to predict conspiracy theory belief, but there are other factors that seem predictive across conspiracy theories. See Uscinski, Joseph, Adam Enders, Amanda Diekman, John Funchion, Casey Klofstad, Sandra Kuebler, Manohar Murthi, Kamal Premaratne, Michelle Seelig, Daniel Verdear, and Stefan Wuchty. 2022. "The Psychological and Political Correlates of Conspiracy Theory Beliefs." Scientific Reports 12: 21672.

9. The predictors of ct belief vary greatly by ct. I think the results by ct should be moved into the text if possible. I think the omnibus measure obscures things. I also think that the result in Fig 2 is likely due to the cts under investigation. That the corporation item is the most engaged with might be driving the extreme liberals to engage more.

10. It seems like neither the left nor the right have a monopoly on conspiracy theorizing in the results on page 12. You might link this finding to Enders, Adam, Christina Farhart, Joanne Miller, Joseph Uscinski, Kyle Saunders, and Hugo Drochon. 2022. "Are Republicans and Conservatives More Likely to Believe Conspiracy Theories?" Political Behavior.

Overall, the paper makes a strong contribution. It should engage with the conspiracy theory literature more - both in motivating the study and interpreting the results. I believe my comments could be handled in an R&R.

Reviewer #1:

R1.1: In this research, the authors investigate the associations between psychological predispositions and support for conspiracy theories on the social media platform X. I was excited to review this paper. It uses a novel methodology to address a problem that is of interest to many social scientists (as well as the public): who engages with conspiracy theories. Explaining real-life social media conspiracy theory engagement is a major strength of the paper. Still, as typical, I had a few questions and comments about the manuscript.

Response: We are glad about the positive evaluation of our work and deeply grateful for the insightful questions and constructive comments received. Engaging with this valuable feedback has significantly strengthened the foundation and clarity of our paper, enriching its overall quality and depth.

R1.2: I admit that, just like the authors, I was perplexed by the non-significant effects observed for conspiracy mentality. As the authors note, this measure has been occasionally criticized for capturing other predispositions, such as levels of political trust, but this does not mean we would expect it to be irrelevant for engagement with conspiracy theories. This almost seems like a sanity check on the dataset, and the fact that this effect was not observed makes me wonder about one major issue in particular: to what degree the study set up might have affected the psychological make up of the final sample.

Participants were asked to grant the researchers access to their X activities through a custom-built app. I would imagine this is more likely to be acceptable for participants who are overall trusting towards scientists or have low levels of conspiracy mentality. Thus, the current results might be more reflective of those who are generally less likely to engage with conspiracy theories. This is consistent with the current results showing that the average likelihood that X engagements supported conspiracy theories was generally low. It would be good to gauge somehow if these levels are comparable to those generally observed on X in the same period among other participants (e.g., those who had public profiles but did not take part in the study). It would also be useful to see if the distribution of variables such as conspiracy mentality in this sample is similar to that observed in other samples.

Response: We appreciate the Reviewer's insightful comment. Like them, we were also intrigued by this unexpected finding. To address the potential of the suggested selection bias influencing our results, we compared the conspiracy mentality mean and distribution observed in our study with the mean scores from a wide range of cultural contexts and a quota-representative U.S. sample in previous research conducted in the same period. Remarkably, the mean score in our study falls well within the cross-cultural range of means and is close to if not even slightly higher than that of a representative sample from the USA. Ultimately, a potential selection bias should presumably exert a comparable influence on the associations with beliefs in false information (from the MIST scale). Nevertheless, this variable showed a statistically significant correlation with support for three conspiracy theories. This observation further reduces the probability of selection bias playing a substantial role in the non-significant finding. We now discuss these points on p. 21:

In our research, subjects were asked to allow access to their X accounts, which might suggest they have a lesser inclination towards conspiracy theories, given their willingness to trust researchers. Contrasting with such a view, the average conspiracy mentality score was notably above the midpoint on the normally-distributed scale ($M = 7.04$, $SD = 1.87$, Skewness = -0.10). Moreover, this mean falls within the general range seen across various studies²² and is slightly higher than the mean in another quota-representative U.S. study conducted in the same period as the present study⁵⁶. Furthermore, it is reasonable to assume that a potential selection bias would have similarly affected the variable of belief in false information, which showed statistically significant associations with three conspiracy theories. Therefore, selection bias does not seem to be a likely explanation for the statistically nonsignificant findings for conspiracy mentality in our study.

Additionally, we now clarify that the disparity between the skewness in specific conspiracy theory support and the normal distribution of conspiracy mentality noted in our study is a frequent occurrence in existing research (see <https://doi.org/10.1016/j.copsyc.2022.101349> for a recent review). Indeed, this divergence has been cited as a fundamental point in the conceptual critique of the conspiracy mentality questionnaire. Accordingly, on p. 21, we now write:

It is further noteworthy that the discrepancy observed in our study between the normal distribution of the scale and the low number of X engagements supporting specific conspiracy theories aligns with common observations in this field of research³⁶.

Indeed, this divergence forms the foundation for conceptual critiques of what the conspiracy mentality questionnaire measures⁵⁷.

R1.3: Furthermore, I appreciated that the model identifying conspiracy theories engagements has been refined, and that the new model “surpassed the untuned model’s performance”. However, the correlations with human evaluations for the fine-tuned model were still modest, especially for certain conspiracy theories.

Response: We express our gratitude to the Reviewer for highlighting this crucial aspect. In an effort to enhance our modelling approach and its validation, we have implemented several measures. Particularly, in response to the feedback provided in comment R.2.7, we have transitioned to utilizing a commercial-grade model with fine-tuned prompts for classification purposes, specifically the latest version of GPT 3.5-turbo, instead of developing our own model. This advanced model demonstrated excellent concordance with the human ratings for four conspiracy theories and substantial concordance for two theories (see Table 8, fourth column). Additionally, we have computed standard performance metrics for the model, which collectively indicate satisfactory performance (see Table 8, fifth through seventh column).

Table 8 | Model Validation for Both the Similarity and Support Classification Models, and Performance Estimates for the Support Classification Model

	Similarity Classification Model		Support Classification Model				
	Human Interrater Reliability α	r Similarity Model Prediction – Average Human Annotation	Mean PABAK: Human Support Annotation	PABAK: GPT – Human Majority Vote	Precision	Recall	F1
1. The coronavirus or the government's response to it is a deliberate strategy to create economic instability or to benefit large corporations over small businesses.	0.90	0.59	0.83	0.93	0.83	0.67	0.74
2. The public is being intentionally misled about the true nature of the Coronavirus, its risks, or the efficacy of certain treatments or prevention methods.	0.90	0.71	0.66	0.75	0.74	0.63	0.68
3. The coronavirus was created intentionally, made by humans, or as a bioweapon.	0.95	0.42	0.89	0.87	0.82	0.61	0.70
4. Politicians or government agencies are intentionally spreading false information, or they have some other motive for the way they are responding to the coronavirus.	0.93	0.65	0.76	0.79	0.59	0.85	0.70
5. The Chinese government intentionally created or spread the coronavirus to harm other countries.	0.94	0.78	0.93	0.91	0.73	0.69	0.71
6. The coronavirus vaccine is either unsafe or part of a larger plot to control people or reduce the population.	0.90	0.77	0.89	0.94	0.80	0.73	0.76

Note. α = Cronbach's alpha. PABAK = Prevalence-Adjusted Bias-Adjusted Kappa. r = Pearson's correlation coefficient.

R1.4: I also admit I found integration of the theory of reasoned action a bit confusing. These variables are based on past work, but I think that the reasoning could be explained a bit better here. Authors predict (and find) that perceived self-efficacy in recognizing misinformation is associated with support for conspiracy theories. This is consistent with past theorising by Khan & Idris (2019) who linked such self-efficacy perceptions to sharing information without verification on social media. But couldn't one equally convincingly argue that perceived self-efficacy in recognizing misinformation could go hand in hand with a lower tendency to share it? For example, Basol et al. (2020; J Cogn) found that playing Bad News (the online fake news game in which players learn about common misinformation techniques) not only improved the ability to spot misinformation techniques but also increased people's level of confidence in their own judgments. Also, given work by Lyons et al. (2021) linking false news susceptibility to **over** confidence in news judgments, it might be important to consider self-efficacy in relation to ability. One idea could be to examine whether there is an interaction effect between the self-efficacy measure and the misinformation susceptibility indices (although I appreciate this is tricky given poor reliability of the latter measures). Overall, I think that the association between perceived self-efficacy in recognizing misinformation and support for conspiracy theories could use a deeper discussion, especially given that it was one of the few significant associations observed in the current study.

Furthermore, it was not clear to me if the effects that were observed in the study were unique to engagement with conspiracy theories or with misinformation more broadly. This might be a nuanced but important distinction, particularly for predictors such as perceived self-efficacy in recognizing misinformation.

Response: We appreciate the Reviewer's suggestion to further examine and discuss the positive association between perceived ability in identifying misinformation and support for conspiracy theories. Although this association with the improved modelling approach now is only observed for one of the six theories, we have attended to these issues in several respects. Initially, referencing the helpful study by Basol et al., we now already in the introduction recognize that some prior research has observed a link between this confidence and an enhanced ability to discern misinformation, see p. 4:

In examining the support for conspiracy theories, the Theory of Reasoned Action²⁹ and its recent adaptations to misinformation processing³⁰ are of relevance. This theory suggests that behavior is influenced by attitudes towards the behavior and normative beliefs. Trust in the authenticity of information on social networks and confidence in identifying false information have been shown to positively correlate, while attitudes towards the importance of validating information have been shown to negatively correlate, with the self-reported spread of misinformation^{30,31}. Whereas this research has focused on the broader topic of misinformation, it has replicated the role of overconfidence with misinformation stimuli that often mapped onto conspiracy theories³². Moreover, recent evidence suggests a correlation between a general overconfidence in one's abilities and belief in conspiracy theories³³. However, contrasting findings exist, such as a study demonstrating that an inoculation intervention enhanced both the ability to discern misinformation (including misinformation related to conspiracy theories) and confidence in judgment, with both constructs being positively correlated³⁴.

Next, we investigated the suggested interaction between confidence and susceptibility to misinformation. However, this relationship did not yield statistical significance, as detailed in the *Supplementary Information*, inserted here for the Reviewer's convenience:

Text S1. Testing for interaction between perceived ability to detect misinformation and self-reported belief in false information

Considering the mixed findings in prior research regarding the role of the perceived ability to spot misinformation, we explored its interaction with belief in false information. It could be hypothesized that this perceived ability might predominantly bolster conspiracy theory support among those already inclined to believe in misinformation. Nonetheless, our analysis revealed that this interaction was not statistically significant when added to any of the models, $\beta_s = 0.01\text{--}0.13$, all $P_{SHolm} = 1.000$.

Additionally, prompted by the Reviewer's insights, we examined if it is overconfidence, rather than confidence itself, that underlies the effects observed, indicative of a quadratic relationship. Yet, this potential quadratic association did not demonstrate statistical significance, as reported in the *Supplementary Information*, inserted here for convenience:

Text S2. Testing for curvilinear influence of perceived ability to detect misinformation

To determine if overconfidence, rather than the perceived ability itself, drives the observed association with conspiracy theory support, we examined the possibility of a curvilinear relationship. These models were tested only controlling for participants followers. However, our analysis revealed that this quadratic association was not statistically significant, $\beta_s = -0.02 \text{ -- } -0.16$, all $P_{SHolm} > 0.460$.

Lastly, in line with the helpful suggestions by the Reviewer and considering our additional findings, we have expanded our discussion on the confidence finding. This includes a detailed examination of the varying outcomes and their meaning in both this study and those involving inoculation, as elaborated on pp. 19-20:

Next, the perceived ability to discern misinformation on social media derived from the theory of reasoned action^{29,30} was associated with statistically significantly elevated levels of support for the conspiracy theory that governments and politicians are intentionally spreading false information. The two additional theory of reasoned

action factors, namely perceived importance of verifying social media information and the belief in the reliability of social media information, showed no statistically significant associations. Recognizing that confidence in one's ability to discern information's veracity is a key resilience factor in psychological inoculation frameworks^{34,52,53}, we conducted further tests to assess the robustness of our finding. Initially, we explored if the association between perceived ability and conspiracy theory support varied with participants' actual misinformation identification skills (i.e., scores on the scale assessing belief in false information; see *Supplementary Information*). However, no statistically significant interaction was found in this regard for any model. We also examined whether overconfidence as reflected in a curvilinear relationship might drive the observed association, but a statistically significant relationship did not emerge (see *Supplementary Information*).

It appears likely that confidence in detecting misinformation, as immediately evaluated in a task-specific inoculation intervention, is distinct from a broader perceived ability to identify misinformation. In an inoculation experiment, participants receive explicit instructions encouraging them to accurately judge the veracity of various types of information. By contrast, this study, which examined behavioral support for conspiracy theories on social media over several years, lacked such direct guidance as it observed naturally-occurring behavior. Furthermore, despite the absence of a statistically significant curvilinear relationship, it is possible that our measure of perceived abilities may still mirror participants' level of overconfidence, consistent with the Dunning-Kruger effect. Building upon prior research and the findings of this study, future investigations should explore whether confidence developed through inoculation interventions over time feeds into or remains distinct from the broader self-perception of abilities evaluated here. This exploration is crucial for helping researchers identify which aspects of digital literacy programs might unintentionally produce adverse effects, like instilling a misguided sense of expertise in detecting misinformation, versus those that cultivate genuinely beneficial confidence.

R1.5: Furthermore, it was not clear to me if the effects that were observed in the study were unique to engagement with conspiracy theories or with misinformation more broadly. This might be a nuanced but important distinction, particularly for predictors such as perceived self-efficacy in recognizing misinformation.

Response: We are grateful to the Reviewer for emphasizing this critical point. Despite perceived self-efficacy now emerging as a less potent predictor with the enhancement of the classification model, we in response to the comment briefly discuss the possibility that the factors identified in the current study might also influence engagement with misinformation. We acknowledge the necessity for future research to directly investigate this possibility. On p. 24 we write:

Future studies are also essential to ascertain whether the outcomes of our investigation can be replicated in the context of behavioral engagement with misinformation, as opposed to conspiracy theories in the online realm. While an exhaustive examination of this possibility exceeds the scope of the current paper, which is explicitly centered on conspiracy theories, the observation that the scale assessing susceptibility to misinformation emerged as a relatively consistent factor implies that analogous mechanisms may be at work.

R1.5: It would also be useful to account for participants' overall engagement with X, for instance by accounting for the number of followers.

Response: We express our gratitude to the Reviewer for this valuable comment. In response, we have meticulously reanalyzed all data, incorporating controls for the number of followers (individuals following the user) and the number of followings (individuals the user follows). Within these models, neither factor exhibited statistical significance.

R1.6: As the authors note, the current methodology “does not facilitate the discernment of causality”, so I would encourage them to be mindful of using causal language across the text (e.g., p. 5 reads: “We proceeded to evaluate the influence of various individual difference factors.”).

Response: We apologize for this linguistic imprecision. We have revised the entire manuscript to avoid language that may imply causality.

R1.7: P. 13 reads: “attributed to men’s higher tendency to subjectively perceive themselves as a marginalized minority group.” – this statement should be supported by a reference.

Response: Following the new analyses, the gender associations became statistically non-significant. Thus, this discussion was omitted from the paper (null findings are not to be discussed according to *Nature Communication's* guidelines).

R1.8: All p-values in the analyses are one-tailed “as predictions were unidirectional”. I would have been more comfortable with this choice if the analytic strategy was pre-registered.

Response: We appreciate the Reviewer’s comment on the use of one-tailed versus two-tailed *P*-tests. In our manuscript, we opted for one-tailed tests based on the specific directional nature of our hypotheses, aligning with *Nature Communications'* statistical reporting guidelines which advocate for flexibility and transparency in this choice (https://www.nature.com/documents/ncomms_statisticalguidance.pdf). Our hypotheses, such as the link between higher misinformation susceptibility and increased support for conspiracy theories, in our opinion necessitate focused tests for their directional predictions. This is supported by literature emphasizing the underutilized yet effective role of one-tailed tests in psychology (<https://daniellakens.blogspot.com/2016/03/one-sided-tests-efficient-and-underused.html>) and generally (<https://www.sciencedirect.com/science/article/pii/S0148296312000550>).

Furthermore, our analysis is already highly conservative, using the Holm correction to control for a high number of 21 tests per model to robustly control for Type I errors. As an example, *P*-values of 0.002 became significant at 0.040 in Model 4. This approach underscores the reliability of our findings. Nevertheless, as we understand that the decision of one- versus two-tailed tests is complex and debated, we have added a paragraph justifying their use on p. 25:

Given that our hypotheses were directional in nature (e.g., predicting a positive association between conspiracy mentality and support for conspiracy theories), we opted for one-tailed tests. While one-tailed tests are sometimes perceived as less conservative compared to two-tailed tests, they provide a more powerful examination of directional predictions, while ensuring a logical alignment with hypotheses⁶⁶. Furthermore, our statistical approach is already inherently conservative, incorporating *P*-value correction for a large number of tests. For instance, the *P*-value of perceived

ability to detect misinformation in Table 4 dropped from 0.002 to 0.040 after this correction.

In summary, we are deeply grateful for the detailed and constructive feedback, which has prompted us to further elaborate and refine various aspects of our paper. We hope that the Reviewer will find these revisions satisfactory.

Reviewer #2:

R2.1: I thank the authors for this important work on how psychological factors relate to the actual support of conspiracy theories on X. Given the prevalence of conspiracy theories online and the concerns about their potential adverse societal outcomes, this work addresses significant and momentous issues. In particular, this work combines surveys with behavioral data to provide empirical evidence of the risk and protective factors related to public endorsement of conspiracy theories. Leveraging a panel of over two thousand X users, subjected to assessment for several conspiracy support factors identified in the literature, the authors analyze how such factors are related to their support for six COVID-19-related conspiracy theories on X, as signified by their posts, replies, likes, and shares. In particular, the authors use machine learning techniques to estimate the dependent variable---the likelihood of supporting each of the six theories. Combining measures of individuals' psychological attitudes and beliefs with observational data in assessing actual support for conspiracy theories online is, to the best of my knowledge, a novel and important addition to the literature. Overall, the manuscript is well-written and easy to follow. Although the analyses appear sound, and the results presented in the manuscript have the potential to inform future research and policy alike, certain methodological details prevent me from fully endorsing its publication in its current form.

Response: We are glad that the Reviewer find our research to make a novel and important addition to the literature. We thank them for providing detailed and constructive feedback, which helped us strengthen our work substantially.

R2.2: ## Construction of the dependent variable

- If I understood correctly, the final dataset used for producing the results in tables 1 and 2 are the ~800k engagements that pass the filter described in the section "Identification of relevant

engagements", rather than the full 12M-strong corpus. This leads me to a few considerations.

- First, I suggest that the authors report the number of users who engage with each of the six theory topics, as users may decide not to engage with any, or to discuss all of them.

Preventing doubts about bias that may be introduced by subsampling engagements could further strengthen the manuscript.

Response: We value the Reviewer's insights and, in acknowledgment of their constructive feedback, wish to delineate that our revised manuscript now encompasses two distinct sets of analyses: (a) The primary analyses, which utilize the entirety of the 7.7 million X engagements dataset (it is noteworthy that this figure is reduced from that of the prior dataset, following the exclusion of duplicate entries), and (b) supplementary analyses that focus exclusively on X engagements with a minimum topical relevance (as indicated by similarity scores exceeding 0.25). The rationale behind our emphasis on the former (a) is its facilitation of the utilization of the complete dataset, thereby enhancing the robustness and generalizability of our findings.

Within the *Supplementary Information*, we now meticulously detail, across Tables S1 through S6, the count of participants whose engagements yielded similarity scores above the 0.25 threshold for the various theories under scrutiny. We provide this information here for the Reviewer's convenience as well:

Theory 1: $N_{Participants} = 2,113$, $N_{Engagements} = 462,452$

Theory 2: $N_{Participants} = 1,993$, $N_{Engagements} = 326,380$

Theory 3: $N_{Participants} = 2,139$, $N_{Engagements} = 369,568$

Theory 4: $N_{Participants} = 2,119$, $N_{Engagements} = 565,274$

Theory 5: $N_{Participants} = 2,015$, $N_{Engagements} = 291,514$

Theory 6: $N_{Participants} = 2,088$, $N_{Engagements} = 381,835$

However, we wish to underscore that a significant proportion of these X engagements, particularly those with lower similarity scores, may not bear direct topical relevance to the theories in question. The adoption of the 0.25 cutoff was the outcome of a data-driven decision-making process. In addressing comment R2.8 (presented below), we substantiated through rigorous validation that this cutoff threshold harbors a high probability of

encompassing all positive instances within a ground truth dataset. Still, most true positives will be found at much higher similarity levels (see Figure S1).

R2.3: Second, the authors could further clarify how to best interpret the dependent variable, which is crucial to assessing the overall contribution. The fact that the sampling universe used to compare the variety of the measure is restricted, the measure itself may not capture the likelihood of a user expressing support for a theory, as most of their expressions are excluded from the analysis. Is the aim of the dependent variable to measure, when users engage with a topic related to a theory, the degree to which they support the theory? This point relates to my suggestions about further unpacking the results of the semantic similarity subtasks.

Response: We express our sincere appreciation to the Reviewer for this comment. As outlined in response to the preceding comment, our primary analyses now focus on the entire dataset, while we also present a summary of the findings from the subset analyses, which involve only X engagements with a similarity score exceeding 0.25 for a specific theory, in the *Supplementary Information*, Text S3, and Tables S1 through S6.

R2.4: Third, to better understand the results it would be useful to know which output of the deBERTa model was used. Given that the model was fine-tuned on a three-class dataset (entail, contradict, neutral), was it just the output related to entailment, or were the two other outcomes also taken into consideration? This potentially changes the meaning of the lower end of the measure (opposition to the theory, neutral stance towards it, irrelevance of the content with respect to the theory, etc.).

Response: We thank the Reviewer for highlighting this essential point for further clarification. After further inspection of our initial models' performance and encouraged by R2.7 (see response to this point for details), we changed our model strategy following their suggestions. Specifically, we now utilize the most recent version of the GPT 3.5 Turbo model (gpt-3.5-turbo-0125) with fine-tuned prompts to identify X engagements that supported a given theory in a binary fashion. As we outline on pp. 34-35, this model produces two outputs: YES (recoded as 1, indicative of support for the theory) and NO (recoded as 0, indicative of absence of support for the theory):

Next, utilizing the most recent OpenAI GPT 3.5 model (gpt-3.5-turbo-0125, temperature = 0), we devised six prompts. Each prompt was designed to ascertain whether a given X engagement expressed support for one of six conspiracy theories. Initially, the prompts stated the respective conspiracy theory under consideration, followed by a comprehensive exposition delineating the criteria for what constituted endorsement or rejection of the theory (see *Supplementary Online Materials* for the full prompts). This delineation was grounded in the preparatory training and guidelines provided to coders prior to the manual annotation of the ground truth dataset. Subsequently, the model was instructed to furnish whether a tweet supported the respective theory in a binary format of “YES” (recoded as 1) and “NO” (recoded as 0).

R2.5: Fourth, on a minor note, clarifying the choices of sampling universe may also further contextualize Figure 1. ~2--15% likelihood of support for each of the six theories is, in absolute terms, very large, and may lead to misunderstanding if quoted out of such context.

Response: We are grateful to the Reviewer for drawing attention to the issue pertaining to R2.3. The updated figure is now based on the full dataset and binary classification. The percentage estimates are now much lower.

R2.6: ## The role of different types of engagement

- Surprisingly, conspiracy mentality seems significantly *negatively* associated with support of conspiracy theories in posts, and similarly, belief in false information appears *negatively* associated with them in replies. I would argue that posts and replies are the most reliable signals, out of the four engagement types, of users' own beliefs in a theory or support for it. If I am reading the SI correctly, I wonder: does this entail that users with a higher conspiracy mentality and lower ability to detect misinformation are less likely to express public support for conspiracy theories on X --- but only if they are initiating a conversation? This is counterintuitive to me but potentially interesting in relation to the use and characteristics of the platform. To this end, relating engagement styles with psychological profiles may be useful to unpack the surprising results.

- More broadly, given the relative heterogeneity of the engagement types and that results may be dominated by more frequent, lower-effort interactions (likes and reposts, which make up three-quarters of the final dataset), I am curious about whether inter-subject differences may

be completely mediated by engagement type, or if conversely, engagement type and individual profiles show structural associations.

Response: We extend our appreciation to the Reviewer for highlighting these significant findings pertaining to the previous version of the manuscript. In alignment with the previous version, we have included these updated results breaking down findings by engagement type in the *Supplementary Information* for the benefit of interested readers. It is important to note that due to computational constraints, we were compelled to estimate these models individually for each category of engagement (that is, 24 models: 6 conspiracy theories each broken down by 4 types of engagement), enabling the models to converge within a reasonable timeframe. This approach was adopted in lieu of directly examining the moderating effects of engagement type, which is computationally highly taxing with data of this size. With the refined modelling approach (engineered prompts via GPT 3.5-turbo, described earlier), the correlations between conspiracy mentality and various types of engagement were rendered statistically non-significant, as detailed in the *Supplementary Information*, Tables S7 through S12. Generally, our analysis revealed that while certain stronger correlations (for example, those involving age) maintained statistical significance for specific theories and engagement types, other correlations became statistically nonsignificant. This shift occurred despite the direction and strength of the beta values aligning with those reported in the analyses involving the full sample, a phenomenon likely attributable to the significantly reduced statistical power. For those readers interested in further investigating these moderating effects, the complete dataset is accessible through the OSF link provided in the manuscript.

R2.7: ## Validation of the machine learning models

- The area where I feel the authors could significantly strengthen the readers' confidence in the presented results is explaining the integration of machine learning models with survey data. I recognize that this intersection is one of the methodological novelties of the manuscript, therefore I think it is important that it is conveyed appropriately. In particular, there is very little discussion about the validation of the outcomes of the machine learning models on two sophisticated tasks: determining whether a piece of content is relevant to the semantic area of a conspiracy theory, and the degree to which it expresses support to the theory.
- First, the description of the methodology would be clearer if the authors could unpack why

the two tasks are necessary. For example, considering that the authors already make use of commercial generative AI for creating samples to finetune the latter model, could they not have used it directly to infer entailment between a contribution and a theory, perhaps to even more accurate results?

Response: We are grateful to the Reviewer for prompting us to elaborate on our step-wise procedure in greater detail. In response to this request, we have expanded the explanation in the methods section of our paper, found on p. 31, to clarify why subsetting the data was deemed a necessary precursor to the training and testing of our model given our resource constraints:

In our pursuit of developing a machine-learning framework adept at discerning and classifying the complex and elusive characteristics of conspiracy theory endorsement, we implemented a semantic similarity search to pinpoint interactions topically related to these theories. This strategy was pivotal from a resource management perspective, as it facilitated a significant reduction in the computational power required for assessing conspiracy theory support in the subsequent phase. The second model, delineated in the ensuing numbered section, demands considerably more resources, as we relied on a commercial-grade model. Estimating the support for conspiracy theories across all 7,7 million X engagements would have substantially surpassed our financial limits. Therefore, this method was essential in enabling the execution of the second, more demanding model within the financial boundaries of our research project.

Subsequently, we now as suggested use the latest gpt-3.5-turbo-0125 model to classify the X engagements in a binary fashion. We explain this procedure on pp. 34-35:

Next, utilizing the most recent OpenAI GPT 3.5 model (gpt-3.5-turbo-0125, temperature = 0), we devised six prompts. Each prompt was designed to ascertain whether a given X engagement expressed support for one of six conspiracy theories. Initially, the prompts stated the respective conspiracy theory under consideration, followed by a comprehensive exposition delineating the criteria for what constituted endorsement or rejection of the theory (see *Supplementary Online Materials* for the full prompts). This delineation was grounded in the preparatory training and

guidelines provided to coders prior to the manual annotation of the ground truth dataset. Subsequently, the model was instructed to furnish whether a tweet supported the respective theory in a binary format of “YES” (recoded as 1) and “NO” (recoded as 0).

R2.8: Second, since the model for semantic similarity appears crucial in determining the dataset on which to infer support for conspiracy theories, it would be important to evaluate its output qualitatively (e.g., does it capture general topics like government intervention in COVID times, or more precisely the issues of government misconduct relevant to the theories?) and quantitatively (e.g., how accurate is it?). As a byproduct, evaluating this model would provide an important description of the dependent variable of the study.

Response: We are grateful for the chance to provide a more detailed account of this methodological approach. In accordance with the request, we have undertaken a more thorough validation of the similarity model. To begin with, we annotated an expanded dataset comprising over 1,000 X engagements with the assistance of three independent raters. Crucially, these engagements were distributed evenly across the six conspiracy theories and were stratified along four distinct similarity score ranges: 0.00-0.25, 0.25-0.50, 0.50-0.75, and 0.75-1.00. The raters were tasked with assigning a score ranging from 1 to 4 to denote the extent to which a particular engagement was thematically related to the corresponding conspiracy theory. We now explain this procedure in more detail on p. 33:

Specifically, we utilized a sentence transformer model (sentence-transformers/all-mpnet-base-v2)⁶⁹ to compute embeddings for each engagement and conspiracy theory, which maps a given text to a 768-dimensional dense vector space. The similarity score is calculated from the distance (squared Euclidean distance) between the engagement vector and the conspiracy theory vector; it ranges from 0 (no resemblance) to 1 (identical). For demonstration, consider the following two posts with relatively high semantic similarity scores regarding the conspiracy theory that vaccines are unsafe and an effort to control and reduce the population (our first theory, see Table 8). The post, “Bill Gates has been talking about population control openly for years – now we have a coronavirus vaccine! Coincidence? I think not!” has an estimated semantic similarity of 0.615, and the post, “Multiple variants of COVID-19 have emerged. Vaccines are our shield. Don’t let conspiracies disarm you.

#FightTheVirus,” an estimated semantic similarity of 0.664. Thus, both posts are correctly identified as topically related to the conspiracy theory. By contrast, the post, “Sunday (tomorrow) is National Ice Cream Day and have we got a gift for you! Join us for an ice cream sundae,” has an estimated semantic similarity score of 0.031, rendering it entirely unrelated. To facilitate efficient search of engagements topically related to the conspiracy theories, we built a FAISS index.⁷⁰

In response to the limitations of our resources, we established a similarity score threshold of 0.25. This decision was strategic and data driven, aiming to exclude completely irrelevant X engagements that would demand significantly higher computing resources, while still maintaining a high level of inclusivity. The threshold was carefully chosen to balance our resource constraints against the risk of Type 2 errors, which involve missing true positives. At this stage, Type 1 errors, which result in including false positives, were not a primary concern as we planned to conduct a detailed analysis of the conspiracy theory support among all X engagements scoring above 0.25 in a subsequent phase with a different model.

To enhance the validity of our methodology, we enlisted three raters to annotate a stratified random sample of 1,019 X engagements, selected from a corpus of millions of tweets gathered from participants’ social networks. This sample was constituted by randomly allocating 50 X engagements for each conspiracy theory across four model similarity score ranges: 0.00-0.25, 0.25-0.50, 0.50-0.75, and 0.75-1.00. The resultant sample size being less than the anticipated 1,200 (calculated from 50 engagements \times 6 conspiracy theories \times 4 similarity brackets) is attributed to the occurrence of fewer than 50 engagements for some theories within the highest similarity bracket of 0.75-1.00. This shortfall was anticipated, as achieving the highest similarity scores necessitates close to linguistic equivalence. The raters evaluated the topical relevance of these engagements in relation to their respective conspiracy theories, employing a four-point scale: 1 signifying “*Topically unrelated*,” 2 “*Likely topically unrelated*,” 3 “*Likely topically related*,” and 4 “*Topically related*.” The interrater reliability (calculated as Cronbach’s alpha) was excellent across each conspiracy theory, and the mean ratings exhibited moderate to strong correlations with the model similarity estimates (see Table 8, first two columns), thereby affirming the model’s validity.”

The interrater reliability, measured using Cronbach’s alpha, was found to be exceptional, as documented in Table 8, first column. Furthermore, a moderate to strong correlation was observed between the similarity scores generated by the model and the average scores from human annotations, see Table 8, second column.

Table 8 | Model Validation for Both the Similarity and Support Classification Models, and Performance Estimates for the Support Classification Model

	Similarity Classification Model		Support Classification Model				
	Human Interrater Reliability α	r Similarity Model Prediction – Average Human Annotation	Mean PABAK: Human Support Annotation	PABAK: GPT – Human Majority Vote	Precision	Recall	F1
1. The coronavirus or the government's response to it is a deliberate strategy to create economic instability or to benefit large corporations over small businesses.	0.90	0.59	0.83	0.93	0.83	0.67	0.74
2. The public is being intentionally misled about the true nature of the Coronavirus, its risks, or the efficacy of certain treatments or prevention methods.	0.90	0.71	0.66	0.75	0.74	0.63	0.68
3. The coronavirus was created intentionally, made by humans, or as a bioweapon.	0.95	0.42	0.89	0.87	0.82	0.61	0.70
4. Politicians or government agencies are intentionally spreading false information, or they have some other motive for the way they are responding to the coronavirus.	0.93	0.65	0.76	0.79	0.59	0.85	0.70
5. The Chinese government intentionally created or spread the coronavirus to harm other countries.	0.94	0.78	0.93	0.91	0.73	0.69	0.71
6. The coronavirus vaccine is either unsafe or part of a larger plot to control people or reduce the population.	0.90	0.77	0.89	0.94	0.80	0.73	0.76

Note. α = Cronbach’s alpha. PABAK = Prevalence-Adjusted Bias-Adjusted Kappa. r = Pearson’s correlation coefficient.

Most critically, additional analyses confirmed that a cutoff threshold of 0.25 was effective in identifying all true positives, as detailed in the *Supplementary Information* (Figure S1), presented here for the Reviewer’s convenience:

Figure S1 | Cumulative Capture of Positives Based on Human Ratings (Majority Vote)

Across Estimated Similarity Scores. For the upper graph more lenient, the majority of raters had to rate an X engagement as “likely topically related” or as “topically related” to the respective conspiracy theory. For the lower more conservative graph, the majority of raters had to rate an X engagement as “topically related” to the respective conspiracy theory. The dashed red line indicates the 0.25 similarity cutoff.

R2.9: Third, there is little description of the validation of the model used to assess support for the theories. This is my main concern with the manuscript in its current form, as this model defines the main dependent variable of the study. The evaluation consists of correlation measures between automated and human labels on ~50--60 samples per theory (of which, a minority are expected to express full support for the theory). Humans The authors could clarify what correlation measure they use to estimate IRR between the five raters, and if they aggregated human judgments when compared with automated measures. Most importantly, especially in light of the low number of validation examples and the out-of-sample data used to finetune the model, it would be important to strengthen confidence in the accuracy of this model. Establishing baselines and testing the model on ground-truth data may be two ways in which the authors could address this concern. I am adding, for reference, some of the existing resources to this end:

Gambini, M., Tardelli, S. & Tesconi, M. The Anatomy of Conspirators: Unveiling Traits using a Comprehensive Twitter Dataset. arXiv (2023).

Langguth, J. et al. COCO: an annotated Twitter dataset of COVID-19 conspiracy theories. *J. Comput. Soc. Sci.* 1–42 (2023) doi:10.1007/s42001-023-00200-3.

Moffitt, J. D., King, C. & Carley, K. M. Hunting Conspiracy Theories During the COVID-19 Pandemic. *Social Media and Society* 7, (2021).

Phillips, S. C., Ng, L. H. X. & Carley, K. M. Hoaxes and Hidden agendas: A Twitter Conspiracy Theory Dataset. in *WWW Companion* (2022). doi:10.1145/3487553.3524665.

Response: We express our appreciation to the Reviewer for their valuable suggestions. In line with their recommendations, we have developed a more extensive ground truth dataset by annotating over 1,000 engagements with the assistance of three raters, close to 200 engagements per theory. In the process of selecting these engagements, those with high similarity scores were deliberately oversampled to ensure an adequate representation of positive instances. This approach was informed by the insights gained from the validation exercises conducted on the similarity model described earlier.

The three raters assessed the engagements in a binary manner to determine whether each supported a specific conspiracy theory. The process resulted in substantial agreement for two of the theories and excellent agreement for the remaining four (see Table 8, third column). Equipped with this enhanced dataset, we reassessed the performance of our initial model and found it to be lacking. Consequently, we transitioned to employing the commercial-grade GPT 3.5 turbo model, as the Reviewer suggested and as previously mentioned. Following this, we classified the ground truth dataset and calculated the PABAK (commonly used in psychology) alongside model performance metrics (which are more commonly used in the computer science field). The PABAK scores were generally substantial and, in most cases, excellent (see Table 8, fourth column). The model performance metrics indicated reasonable effectiveness, considering the complexity of the task at hand (see Table 8, fifth through seventh column).

In the manuscript, we explain this procedure on pp. 34-35:

For the subsequent evaluation of the support classification model, we initially created a ground truth test dataset. Specifically, three evaluators annotated a series of engagements extracted from the X engagement within the social networks of participants, rather than the participants themselves. We selected 200 engagements per conspiracy theory for this purpose. Notably, considering the prior validation of the similarity model, which demonstrated that a majority of positive instances were identified at higher levels of similarity (see *Supplementary Information*), we intentionally oversampled engagements exhibiting higher similarity scores. For each conspiracy theory, we included 25% of engagements with similarity scores ranging from 0.25 to 0.45, another 25% with scores from 0.45 to 0.65, and the remaining 50% with scores exceeding 0.65. This approach yielded a total of 1136 X engagements after the exclusion of tweets in other languages than English. The agreement between raters was substantial for two theories and excellent for four theories (see Table 8, third column).

Next, utilizing the most recent OpenAI GPT 3.5 model (gpt-3.5-turbo-0125, temperature = 0), we devised six prompts. Each prompt was designed to ascertain whether a given X engagement expressed support for one of six conspiracy theories. Initially, the prompts stated the respective conspiracy theory under consideration, followed by a comprehensive exposition delineating the criteria for what constituted

endorsement or rejection of the theory (see *Supplementary Online Materials* for the full prompts). This delineation was grounded in the preparatory training and guidelines provided to coders prior to the manual annotation of the ground truth dataset. Subsequently, the model was instructed to furnish whether a tweet supported the respective theory in a binary format of “YES” (recoded as 1) and “NO” (recoded as 0).

Next, we undertook a multifaceted validation of the model prompts. Initially, we assessed the concordance between the GPT model’s responses and the ground truth data. This phase of validation revealed excellent agreement between machine and human evaluations for four of the conspiracy theories, and substantial agreement for the remaining two (refer to Table 8, fourth column). Following this, we appraised the performance metrics of the model. The precision metrics were deemed satisfactory, and the recall rates were considered acceptable for all theories except one. The exception was the fourth conspiracy theory. However, whereas it exhibited a relatively lower precision metric, the recall was high, compensating for that imprecision. Crucially, across all theories, the F1 scores were either closely approaching or surpassing 0.7, indicative of decent model performance in a task as nuanced as discerning support for conspiracy theories within brief X engagements.

R2.10: On a minor note, the authors state that studies that endeavored to identify determinants of engagement with conspiracy theories have rarely made use of large-scale behavioral insights. While I agree that the combination of survey measures and large-scale analyses in this field is rare, there are several studies addressing precursors to engagement with online conspiracy theories or support for them, such as:

Nera, K., Leveaux, S. & Klein, P. P. L. E. A “Conspiracy Theory” Conspiracy? A Mixed Methods Investigation of Laypeople’s Rejection (and Acceptance) of a Controversial Label. *Int. Rev. Soc. Psychol.* 33, (2020).

Phadke, S., Samory, M. & Mitra, T. What Makes People Join Conspiracy Communities?: Role of Social Factors in Conspiracy Engagement. *Proceedings of the ACM on Human-Computer Interaction CSCW*, (2020).

Response: We express our gratitude to the Reviewer for highlighting these significant studies. We acknowledge that the initial phrasing regarding this matter was not precise enough. Rather than suggesting a dearth of research utilizing large-scale behavioral data to explore support for conspiracy theories, our intention was to underscore the scarcity of studies that merge this approach with comprehensive psychological data. We have now refined our wording to convey this point more accurately and have also incorporated a reference to the noteworthy study by Phadke et al. (2020), see pp. 2-3:

Yet, while numerous studies have endeavored to identify these determinants¹⁴, research efforts have been predominantly constrained by an over-reliance on self-reported data and very rarely combined them with large-scale behavioral insights from social media platforms. Conversely, whereas certain studies offer compelling analyses of large-scale online behavior related to conspiracy theories on social media platforms¹⁵, they typically fail to integrate these observations with in-depth psychological insights, which are challenging to gauge accurately without the use of surveys. Owing to the significant inconsistencies between self-reported intentions and actual behaviors¹⁶, our current understanding of the psychological factors that correlate with online support for conspiracy theories as documented in big data remains limited.

Additionally, in the discussion surrounding network effects, we present a more detailed description of the Phadke et al. study, its findings, and methodological approach. Juxtaposing it with the aspects from the current study, we suggest how future research combining aspects and insights from both studies could provide novel insights, see p. 23:

Further, the individuals that users choose to follow and interact with can also account for variations in conspiracy theory support, as the dissemination of such content often hinges on the relationship with and trust in the individual sharing the information⁶¹. For instance, a recent study provides an intriguing perspective, tracking a substantial number of Reddit users longitudinally¹⁵. Initial interactions with individual conspiracy theorists and subsequent active recruitment by these individuals often preceded and led to increased participation in conspiracy theory groups. Additionally, ostracization from groups engaged in more conventional topics acted as a driving force toward such engagement. While Reddit's distinct group structure is not directly comparable to that

of the X platform, employing longitudinal network analyses that combine survey and behavioral data from social media could be instrumental in understanding how changes in individual social networks over time are influenced by individual characteristics and vice versa. This approach could be particularly effective in mapping the temporal trajectories and evolution of individuals who exhibit the risk factors identified in our study.

Additionally, the other study mentioned by the Reviewer, which investigates individuals' reactions to the term "conspiracy theory," is pertinent to our analysis of the strengths and weaknesses of the conspiracy mindset scale. We have incorporated this study into our discussion and elaborated on it on p. 20, as follows:

The absence of a statistically significant finding with the conspiracy mentality scale³⁵ is somewhat perplexing. Unlike other scales that directly measure beliefs in a set of conspiracy theories, this tool gauges a general propensity towards such beliefs. One of the strengths of this approach is its avoidance of direct references to terms like "conspiracy theory." Such terminology is often objected to by actual believers in conspiracy theories and even perceived as part of a conspiracy itself⁵⁴, potentially disrupting accurate measurement.

In conclusion, we extend our sincere thanks to the Reviewer for their insightful and constructive feedback. We are confident that addressing their comments has substantially enhanced the quality of our manuscript, and we hope that the Reviewer is satisfied with the revisions we have implemented.

Reviewer #3:

R3.1: It isn't clear how causal conspiracy theories, or beliefs in conspiracy theories are. I am sure they are sometimes, but not all of the time. The authors should be more circumspect about their role in driving outcomes. See for example: Uscinski, Joseph, Adam M. Enders, Casey Klofstad, and Justin Stoler. 2022. "Cause and Effect: On the Antecedents and Consequences of Conspiracy Theory Beliefs." *Current Opinion in Psychology* 101364.

Response: We express our gratitude to the Reviewer for bringing attention to this crucial aspect. We concur with the concerns regarding the tendency, both in this and other fields, to deduce causality and social significance merely from correlations. Consequently, we have made significant additions to our paper. First, already in the introduction, we refer to the significant paper noted by the Reviewer to provide more nuance, see p. 2:

Empirical research swiftly documented the repercussions of these conspiracy theories on global public health. Beliefs in the theories were not only correlated with, but also prospectively predicted decreased compliance with preventive measures and the increased likelihood of resorting to potentially dangerous alternative treatments^{1,9,10} (but see ¹¹ for a critical discussion of the assumed consequences of conspiracy theories).

Second, we expanded our discussion on the limitations pertaining to causal inferences and social consequences. This includes referencing the paper and applying its arguments. As elaborated on pp. 24-25, we state:

Whereas our dataset addresses a significant gap in the literature by incorporating behavioral outcomes, it does not facilitate the discernment of causality. It is conceivable that the predictors under study influenced conspiracy theory support on social media; however, the inverse likely also holds true. For instance, individuals with politically extreme orientations may propagate and seek out conspiracy theories online that align with their views, further reinforcing their political extremity. This potential feedback mechanism warrants exploration in future research.

It is also crucial to acknowledge that the extent to which beliefs in conspiracy theories have significant consequences remains a contentious subject. A primary critique centers on the problematic extrapolation of causality from mere correlations¹¹. While certain experimental and longitudinal studies indicate potential downstream impacts of conspiracy theories on social orientations and behavioral intentions^{64,65}, recent meta-analytic cross-lagged research has revealed only modest longitudinal effects, in this case in the context of health-related responses¹. Our study is unable to investigate the downstream effects of endorsing conspiracy theories online in various life domains. However, the prevailing assumption of causality,

predominantly derived from correlational studies, highlights this as a significant issue warranting further exploration¹¹.

R3.2: Page 3. Conspiracy theories are not always "intricate". The authors should settle on a published definition and cite it on the top of page 3.

Response: We apologize for this imprecision. We have now omitted the word "intricate" and cite a succinct definition, see p. 2:

Conspiracy theories can be succinctly defined as beliefs about "the machinations of a small group of powerful people, working in secret, against the common good"⁵ (p. 151). Thus, although conspiracy theories evolve continuously and track important societal events, they usually represent shared features (e.g., abuse of power, deception)⁶. Importantly, while these theories are often bereft of credible evidence, they are not necessarily based fully on falsehoods⁷.

R3.3: Page 3. There is a rich literature on conspiracy theories and the factors associated with belief in them. You might want to remove the "misinformation studies" section on page 3 and focus on conspiracy theory studies. This current study seems to be about conspiracy theories, so it makes sense to motivate it from that rich literature. See for example:

Brandenstein, Nils. 2022. "Going Beyond Simplicity: Using Machine Learning to Predict Belief in Conspiracy Theories." *European Journal of Social Psychology* 52: 910-30.

Uscinski, Joseph, Adam Enders, Amanda Diekman, John Funchion, Casey Klofstad, Sandra Kuebler, Manohar Murthi, Kamal Premaratne, Michelle Seelig, Daniel Verdier, and Stefan Wuchty. 2022. "The Psychological and Political Correlates of Conspiracy Theory Beliefs." *Scientific Reports* 12: 21672.

It comes off odd that the authors ignore much of the conspiracy theory literature and then say that because the misinformation literature is not about conspiracy theories, there is a need for the present study (bottom of page 3). I think the authors should dump much of the misinformation literature and motivate this study from the ct literature since that is what this study is about.

This same issue comes up again on page 4 where the authors claim that previous research (cites 23 and 24) show that republicans engage more with conspiracy theories. But, as the authors already said, these studies are about untrustworthy websites and misinformation, not conspiracy theories specifically.

Response: We express our profound gratitude to the Reviewer for their valuable recommendations, which have significantly enhanced the alignment of our introduction and literature review with the specific focus of our study on conspiracy theories, as opposed to a broader emphasis on misinformation. In response, we have extensively revised the introduction, primarily by omitting numerous studies related to misinformation, as per the Reviewer's advice. We have also incorporated most of the research highlighted by the Reviewer, along with other pertinent studies. It is important to acknowledge that the boundary between misinformation and conspiracy theory research is often indistinct, with many studies on misinformation frequently employing content that pertains to conspiracy theories. Therefore, we have selectively retained such studies where they provide relevant insights, while ensuring greater clarity about this overlap in our revised manuscript. Furthermore, the section discussing the relationship between political orientation, affiliation, and conspiracy theories has been thoroughly reworked to focus primarily on research pertaining to conspiracy theories, incorporating the helpful works recommended by the Reviewer. The revised section can now be found on pp. 3-5 of our manuscript and reads as follows:

Nevertheless, the insights gained from existing research offer valuable guidance for studies that aim to integrate both sources of data. Numerous self-report studies have adopted an approach of investigating individual differences. Notably, narcissism has consistently shown a strong association with conspiracy beliefs, as corroborated by meta-analytic evidence besides others^{14,17}. The allure of conspiracy theories for narcissistic individuals may lie in their potential to garner attention, affirm their perceived superiority, and display their unique insights or viewpoints. Moreover, individual differences in denialism and a need for chaos seem to play an important role. Denialism, characterized by the rejection of expert narratives and the tendency to seek unconventional explanations for significant events¹⁸, has been consistently linked to beliefs in COVID-19 conspiracy theories in self-report research¹⁹. By contrast, the need for chaos, a desire to provoke or exacerbate disorder, has been identified as a key motivator behind the spread of hostile political rumors^{20,21}.

Individuals' more general political alignment is another crucial factor in understanding their conspiracy theory beliefs. Extensive cross-cultural research²² indicates that individuals at both ends of the political spectrum are more likely to endorse conspiracy theories^{23,24}. Indeed, individuals on both the left and right harbor suspicions that the opposite side engages in conspiracies²⁵. Behavioral data, such as user interactions on social media, further support these findings, showing that polarized individuals are more engaged in propagating information related to conspiracy theories²⁶. This trend of political extremity underlying conspiracy theory beliefs is particularly pronounced when theories concern powerful entities²⁷. Nonetheless, right-leaning individuals show a greater tendency towards conspiracy theories targeting marginalized groups²⁷. Certain conservative movements, as for instance exemplified by voting for Trump compared to Biden, still seem to exhibit a stronger overall inclination towards conspiracy beliefs²⁸.

In examining the support for conspiracy theories, the Theory of Reasoned Action²⁹ and its recent adaptations to misinformation processing³⁰ are of relevance. This theory suggests that behavior is influenced by attitudes towards the behavior and normative beliefs. Trust in the authenticity of information on social networks and confidence in identifying false information have been shown to positively correlate, while attitudes towards the importance of validating information have been shown to negatively correlate, with the self-reported spread of misinformation^{30,31}. Whereas this research has focused on the broader topic of misinformation, it has replicated the role of overconfidence with misinformation stimuli that often mapped onto conspiracy theories³². Moreover, recent evidence suggests a correlation between a general overconfidence in one's abilities and belief in conspiracy theories³³. However, contrasting findings exist, such as a study demonstrating that an inoculation intervention enhanced both the ability to discern misinformation (including misinformation related to conspiracy theories) and confidence in judgment, with both constructs being positively correlated³⁴.

Rather than focusing on identifying related predisposing factors, certain studies have endeavored to directly measure individuals' susceptibility to believing in conspiracy theories or misinformation more broadly through the use of specialized psychometric scales. Arguably most influential has been the conspiracy mentality questionnaire³⁵, thought to capture a disposition towards conspiratorial viewpoints across various subjects³⁶. While the scale was originally conceptualized as distinct

from specific conspiracy theory beliefs, it has consistently demonstrated a positive correlation with many of these beliefs³⁶. Additionally, a recently developed scale assesses susceptibility to misinformation, by measuring the propensity to accept politically-balanced false information, including conspiratorial content, and the tendency to reject accurate information³⁷. Of particular relevance to the present research, higher misinformation susceptibility as indexed by the scale was positively associated with COVID-19 conspiracy beliefs³⁸.

Although the research landscape on conspiracy theories is developing quickly, it is still marked by diverse and sometimes conflicting perspectives. For instance, while some studies highlight the sizeable explanatory power of personality factors²⁸, recent data-driven research suggests that distrust in governments (closely related to denialism) and perceptions of powerful others (closely related to conspiracy mentality) may have more explanatory power than personality factors³⁹. However, arguably one of the most critical limitations in existing research is its primary reliance on self-report data, which hinders precise identification of individuals and populations behaviorally engaged in or resistant to conspiracy beliefs. This gap underscores the need for research testing the role of theoretically-based factors with large-scale and naturalistic behavioral data to inform targeted interventions.

R3.4: Some good studies of ct engagement to look at are:

Bessi, Alessandro, Mauro Coletto, George Alexandru Davidescu, Antonio Scala, Guido Caldarelli, and Walter Quattrociocchi. 2015. "Science Vs Conspiracy: Collective Narratives in the Age of Misinformation." *PLoS ONE* 10: e0118093.

Mancosu, Moreno, and Federico Vegetti. 2020. "'Is It the Message or the Messenger?': Conspiracy Endorsement and Media Sources." *Social Science Computer Review*.

Miani, Alessandro, Thomas Hills, and Adrian Bangerter. 2021. "Loco: The 88-Million-Word Language of Conspiracy Corpus." *Behavior Research Methods*.

Response: We extend our gratitude to the Reviewer for bringing attention to these enlightening studies. The study conducted by Bessi et al. (2015) has been integrated into the

above review of the literature. For ease of reference, we reiterate the pertinent sentence from our review (p. 4):

Behavioral data, such as user interactions on social media, further support these findings, showing that polarized individuals are more engaged in propagating information related to conspiracy theories²⁶.

Moreover, the study by Mancosu and Vegetti is now discussed on p. 23:

In addition to the architecture of social media platforms, the origin of the information, beyond the identity of the individual disseminating it, is influential. For example, research indicates that individuals who subscribe to conspiracy theories demonstrate a higher propensity to trust information originating from alternative sources as opposed to mainstream outlets⁶². This implies that the users' connection to the primary source of information, such as a news website, exerts an impact that could beneficially broaden the methodological framework employed in the current study.

Finally, the study by Miani et al. (2021) is now cited at the beginning of the introduction, when defining the term "conspiracy theory," see p. 2:

Thus, although conspiracy theories evolve continuously and track important societal events, they usually represent shared features (e.g., abuse of power, deception)⁶.

R3.5: The paper glosses over the data collection effort very quickly. It seems like the 2300 users averaged 5000 interactions with these six conspiracy theories. That seems like a lot. Could the authors explain this a bit? Are these 2272 people representative of the US? Can we generalize from this sample? Were they picked because they are already known to be conspiratorial?

Response: We appreciate the Reviewer's suggestion to provide more detail on our data collection methods. In response, we have revised the manuscript to address each of the three points raised. Firstly, concerning the number of interactions per user, these figures should be understood in the context of our methodological approach. We have elaborated on this in the revised manuscript on pp. 33-34:

Using this cutoff, we identified the following relevant X engagements among the 7,7 million engagements with similarity scores above 0.25 for at least one conspiracy theory, which were subjected to the support classification model presented below: 462,452 engagements for Theory 1, 326,380 for Theory 2, 369,568 for Theory 3, 565,274 for Theory 4, 291,514 for Theory 5, and 381,835 for Theory 6. This proportion of the 7,7 million overall participant engagements on the platform during the COVID-19 pandemic might first appear substantial. However, the similarity cutoff was deliberately set to be inclusive, to minimize Type-2 errors in detection. As displayed in Figure 1, the numbers of actual positives were much smaller. Thus, the vast majority of engagements with a similarity score above 0.25 do neither support (see Figure 1) nor concern a given conspiracy theory (see *Supplementary Information*, Figure S1).

Indeed, while being inclusive, a similarity score above 0.25 indicates minimal relevance to the respective conspiracy theory as shown in Figure S1:

Figure S1 | Cumulative Capture of Positives Based on Human Ratings (Majority Vote)

Across Estimated Similarity Scores. For the upper graph more lenient, the majority of raters had to rate an X engagement as “likely topically related” or as “topically related” to the respective conspiracy theory. For the lower more conservative graph, the majority of raters had to rate an X engagement as “topically related” to the respective conspiracy theory. The dashed red line indicates the 0.25 similarity cutoff.

Secondly, concerning our sample, we discovered an oversight in the initial demographic presentation, where the data was inadvertently weighted by participants' different numbers of X engagements. This oversight initially led to unexpected deviations from the representativeness criteria set by our online panel provider. We have rectified this by offering corrected demographic estimates, alongside a new column in the updated table. This column draws comparisons with the U.S. Twitter population based on a PEW study. The revised data reveals that our sample closely aligns with many key demographic characteristics, though it is important to note a slight overrepresentation of women, Democrats, and Republicans. These details, along with other pertinent information, are now thoroughly documented in the updated table and accompanying analysis, see p. 26:

Using the survey company CloudResearch, we recruited a sample of 2,506 U.S. Americans that was intended to be representative. Participant demographics are presented in Table 7. While the sample approached representativeness of the U.S. Twitter population in term of age, employment status, civil status, education, and race, participants identifying as women, as Democrats, or as Republicans were slightly overrepresented. However, regarding gender representation, it is crucial to note that our study included a third gender option, which was not considered in the PEW report on the U.S. Twitter population. This additional category in our methodology could explain some of the observed differences in gender distribution when compared to the PEW findings.

Table 7 Participant Demographics		
	Study Sample	U.S. Twitter users ⁶⁷
Age M (SD)	41.78 (14.87)	39.90 (14.88)
Gender in %		
Men	45.8	50.35
Women	52.0	49.65
Third gender / nonbinary / other	2.10	NA
Employment in %		
Working full-time	48.80	68.62 (combined)
Working part-time	12.60	
Unemployed and looking for work	8.82	9.35
A homemaker or stay-at-home parent	7.50	NA
Student	5.75	NA
Retired	10.90	7.94
Other	5.47	7.31
Civil Status in %		
Married	33.60	41.62
Living with a partner	12.40	8.07
Widowed	3.07	2.27
Divorced/Separated	10.60	9.60
Never been married	38.40	38.44
Education in %		
Less than high school degree	1.36	3.72
High school graduate (high school diploma or equivalent including GED)	17.70	20.46
Some college but no degree	26.00	23.32
Associate degree in college (2-year)	12.90	10.19
Bachelor's degree in college (4-year)	28.90	26.35
Master's degree	10.50	12.52
Doctoral degree	1.00	3.43 (combined)
Professional degree (JD, MD)	1.48	
Political Affiliation in %		
Republican	24.80	20.88
Democrat	41.10	35.74
Independent	28.80	29.05
Something else	5.11	12.62
Race in %		
White	77.60	77.58
Black or African American	12.40	11.32
American Indian or Alaska Native	1.36	NA
Asian	4.27	NA
Native Hawaiian or Pacific Islander	0.32	NA
Other	4.07	5.93

In the discussion, we highlight the slight overrepresentation of some demographics, see p. 24:

Although the sample showed relatively close resemblance of many core demographic variables of the U.S. population on X, it was not fully representative, somewhat limiting its generalizability. In particular, women, Democrats, and Republicans were slightly overrepresented.

Finally, in response to Reviewer 1's comment (R1.2), as now detailed in our manuscript, it is also noteworthy that our sample exhibited a level of conspiratorial thinking comparable to that found in a recent, representative survey conducted within the same timeframe as our study. This parallel in conspiratorial thought patterns is elaborated upon on p. 21:

In our research, subjects were asked to allow access to their X accounts, which might suggest they have a lesser inclination towards conspiracy theories, given their willingness to trust researchers. Contrasting with such a view, the average conspiracy mentality score was notably above the midpoint on the normally-distributed scale ($M = 7.04$, $SD = 1.87$, Skewness = -0.10). Moreover, this mean falls within the general range seen across various studies²² and is slightly higher than the mean in another quota-representative U.S. study conducted in the same period as the present study⁵⁶.

R3.6: The study focuses on a handful of traits that have been found to predict conspiracy theory belief, but there are other factors that seem predictive across conspiracy theories. See Uscinski, Joseph, Adam Enders, Amanda Diekman, John Funchion, Casey Klofstad, Sandra Kuebler, Manohar Murthi, Kamal Premaratne, Michelle Seelig, Daniel Verdier, and Stefan Wuchty. 2022. "The Psychological and Political Correlates of Conspiracy Theory Beliefs." *Scientific Reports* 12: 21672.

Response: We express our gratitude to the Reviewer for bringing attention to this pivotal study. It has been referenced in the revised literature review within the introduction of our paper. Additionally, we discuss the study's findings, focusing on variables that could be relevant for subsequent research building upon our results. These discussions are detailed on p. 22:

It remains important to recognize that the overall variance explained in our sample was relatively low, ranging from 7% to 22%. This observation underscores the idea

that factors other than those measured in this research may play a role. Whereas we attempted to include a broad range of the factors previously proposed to predict conspiracy beliefs, it was not feasible to include every possible variable. In future research, it may be beneficial to test the variables recently proposed in a meta-analysis on conspiracy beliefs¹⁴, which was published after the participant-level data were collected for this study. In a similar vein, a recent study conducted in the United States, which examined a broad range of factors, suggested that variables such as populism, Manicheanism, and alignment with specific political figures (for example, Donald Trump) might be important²⁸.

R3.7: The predictors of ct belief vary greatly by ct. I think the results by ct should be moved into the text if possible. I think the omnibus measure obscures things. I also think that the result in Fig 2 is likely due to the cts under investigation. That the corporation item is the most engaged with might be driving the extreme liberals to engage more.

Response: As suggested, we now present separate analyses for each conspiracy theory as we agree with the Reviewer's assessment. Moreover, we discuss the appeal of different theories to people with different political orientations pp. 17-18:

Individuals who exhibit politically extreme tendencies, regardless of whether they lean towards the left or right, demonstrated a statistically significantly elevated behavioral support for three out of six conspiracy theories. This outcome coherently aligns with findings from a substantial, cross-cultural, self-report study that identified elevated levels of conspiracy beliefs among those situated at the political extremes²². It also reinforces the results from extensive comparative studies, suggesting that neither side has a monopoly on conspiracy theorizing²⁵. Notably, our study provides a crucial expansion to existing research by positing that this phenomenon is also intrinsic to the dissemination of conspiracy theories on online social media platforms. Contradicting the view that particularly conservatives are chiefly responsible for the online proliferation of conspiracy theories^{42,49-51}, our findings imply that interventions aiming to curb the spread of such theories should strategically target individuals on both ends of the political spectrum. However, even within these associations, nuances exist. The conspiracy theory suggesting that the virus and the response to it are part of

an effort to benefit large corporations at the expense of small businesses found particularly strong resonance among those at the far left of the political spectrum. Such theories, which emphasize economic exploitation, seem to align with left-wing individuals' concern for equality. By contrast, the notion that the public is deliberately being deceived about the true nature of the virus and its prevention found the most support among those on the extreme right. Of note, for the theory positing that the virus is human-made and a bioweapon, political orientation displayed a linear association with the propensity to behaviorally endorse the theory, suggesting that individuals with right-leaning political views may be particularly susceptible. However, the conspiracy theory that governments and politicians are purposefully disseminating false information appealed comparably to both ends of the political spectrum, highlighting its significance in appealing to extreme individuals across the political divide²⁷.

R3.8: It seems like neither the left nor the right have a monopoly on conspiracy theorizing in the results on page 12. You might link this finding to Enders, Adam, Christina Farhart, Joanne Miller, Joseph Uscinski, Kyle Saunders, and Hugo Drochon. 2022. "Are Republicans and Conservatives More Likely to Believe Conspiracy Theories?" *Political Behavior*.

Response: We extend our gratitude to the Reviewer for recommending this insightful research, which aided in further contextualizing our findings. In accordance with the suggestion, we have incorporated this study into the revised literature review, see pp. 3-4:

Individuals' more general political alignment is another crucial factor in understanding their conspiracy theory beliefs. Extensive cross-cultural research²² indicates that individuals at both ends of the political spectrum are more likely to endorse conspiracy theories^{23,24}. Indeed, individuals on both the left and right harbor suspicions that the opposite side engages in conspiracies²⁵. Behavioral data, such as user interactions on social media, further support these findings, showing that polarized individuals are more engaged in propagating information related to conspiracy theories²⁶. This trend of political extremity underlying conspiracy theory beliefs is particularly pronounced when theories concern powerful entities²⁷. Nonetheless, right-leaning individuals show a greater tendency towards conspiracy theories targeting marginalized groups²⁷. Certain conservative movements, as for

instance exemplified by voting for Trump compared to Biden, still seem to exhibit a stronger overall inclination towards conspiracy beliefs²⁸.

We have also implemented it into the discussion section, as detailed on p. 17:

Individuals who exhibit politically extreme tendencies, regardless of whether they lean towards the left or right, demonstrated a statistically significantly elevated behavioral support for three out of six conspiracy theories. This outcome coherently aligns with findings from a substantial, cross-cultural, self-report study that identified elevated levels of conspiracy beliefs among those situated at the political extremes²². It also reinforces the results from extensive comparative studies, suggesting that neither side has a monopoly on conspiracy theorizing²⁵. Notably, our study provides a crucial expansion to existing research by positing that this phenomenon is also intrinsic to the dissemination of conspiracy theories on online social media platforms.

R3.9: Overall, the paper makes a strong contribution. It should engage with the conspiracy theory literature more - both in motivating the study and interpreting the results. I believe my comments could be handled in an R&R.

Response: We are immensely pleased to learn that the Reviewer recognizes our paper as making a substantial contribution. We sincerely appreciate the detailed and constructive feedback, which has been instrumental in enhancing the framing and contextualization of our research. This input has also been invaluable in strengthening our interpretations and conclusions.

REVIEWERS' COMMENTS

Reviewer #1 (Remarks to the Author):

I was happy to see a revision of this manuscript. In my current reading, I focused on the issues I had raised in my previous review. Authors addressed most of my concerns. I was a bit disappointed not to see an empirical attempt to determine whether the effects observed here were unique to engagement with conspiracy theories (versus with misinformation), but I also think it is reasonable for the authors to argue this is beyond the scope of the current project.

Reviewer #2 (Remarks to the Author):

I thank the authors for their effort towards improving the manuscript and corroborating the results: I find their resubmitted paper fundamentally improved in both respects. I consider the main critiques that I raised about the former iteration of the manuscript to be satisfactorily addressed, which increases my confidence in the current iteration. I appreciate the authors' thoughtful response to my and the other reviewers' suggestions.

In particular, I thank the authors for substantially reworking and thoroughly validating their analytical pipeline. The authors deduplicated the dataset which, although resulting in a relatively reduced overall number of observations, is likely to yield more accurate estimates. Most importantly, the all-new model for inferring support for the six conspiracy theories has been convincingly presented as correlating with human assessments: while the results depart somewhat from the first iteration of the manuscript, the current iteration appears better performing and more interpretable. I also think that the authors improved the manuscript by clarifying the definition and operationalizations of the outcome variable as well as by including supplementary analyses testing potential differences between different types of engagement on X.

I am still somewhat puzzled by the lack of significance about the CMQ as a regressor, considering the similarities between the items in the questionnaire and the conspiracy theories under study, such as "I think that politicians usually do not tell us the true motives for their decisions" and "governments and politicians intentionally spread misinformation". Yet, I understand and accept the authors' critique of the questionnaire itself. In case the authors wish to further investigate this matter, though I do not think it necessary given the current status of the paper and its already rich set of analyses, one avenue may be to consider users' responses to individual items of the questionnaire, to test whether the lack of significance may be the result of only some sub-aspect of generic conspiracy belief being relevant to a specific theory. Conversely, the authors may investigate how similarities in the conspiracy theories relate to similarly significant regressors, following recent findings that show that correlates of belief in conspiracy theories may vary, e.g., depending on the nature of the theories, such as depending on whether they are more specific/event-based or more generic.

Bowes, S. M., Costello, T. H. & Tasimi, A. The Conspiratorial Mind: A Meta-Analytic Review of Motivational and Personological Correlates. *Psychol. Bull.* 149, 259–293 (2023).

Finally (although within the scope of the present work), I agree with the authors that understanding how changes in individual social networks over time are influenced by individual characteristics would be an intriguing question, as there is limited empirical evidence on both the evolution of individual conspiracy beliefs and their relationship with a potentially corresponding evolution in believers' social connections.

Reviewer #2 (Remarks on code availability):

My knowledge of R is limited, therefore I will not comment on the correctness of the code. The code is readable and commented so that its main functions are recognizable and understandable.

Reviewer #3 (Remarks to the Author):

The authors have addressed my concerns.

Reviewer 2:

R2.1: I thank the authors for their effort towards improving the manuscript and corroborating the results: I find their resubmitted paper fundamentally improved in both respects. I consider the main critiques that I raised about the former iteration of the manuscript to be satisfactorily addressed, which increases my confidence in the current iteration. I appreciate the authors' thoughtful response to my and the other reviewers' suggestions.

In particular, I thank the authors for substantially reworking and thoroughly validating their analytical pipeline. The authors deduplicated the dataset which, although resulting in a relatively reduced overall number of observations, is likely to yield more accurate estimates. Most importantly, the all-new model for inferring support for the six conspiracy theories has been convincingly presented as correlating with human assessments: while the results depart somewhat from the first iteration of the manuscript, the current iteration appears better performing and more interpretable. I also think that the authors improved the manuscript by clarifying the definition and operationalizations of the outcome variable as well as by including supplementary analyses testing potential differences between different types of engagement on X.

I am still somewhat puzzled by the lack of significance about the CMQ as a regressor, considering the similarities between the items in the questionnaire and the conspiracy theories under study, such as "I think that politicians usually do not tell us the true motives for their decisions" and "governments and politicians intentionally spread misinformation". Yet, I understand and accept the authors' critique of the questionnaire itself. In case the authors wish to further investigate this matter, though I do not think it necessary given the current status of the paper and its already rich set of analyses, one avenue may be to consider users' responses to individual items of the questionnaire, to test whether the lack of significance may be the result of only some sub-aspect of generic conspiracy belief being relevant to a specific theory. Conversely, the authors may investigate how similarities in the conspiracy theories relate to similarly significant regressors, following recent findings that show that correlates of belief in conspiracy theories may vary, e.g., depending on the nature of the theories, such as depending on whether they are more specific/event-based or more generic.

Bowes, S. M., Costello, T. H. & Tasimi, A. The Conspiratorial Mind: A Meta-Analytic Review of Motivational and Personological Correlates. *Psychol. Bull.* 149, 259–293 (2023).

Finally (although within the scope of the present work), I agree with the authors that understanding how changes in individual social networks over time are influenced by individual characteristics would be an intriguing question, as there is limited empirical evidence on both the evolution of individual conspiracy beliefs and their relationship with a potentially corresponding evolution in believers' social connections.

Response: Reviewer 2 states that if we “wish to further investigate the matter” of the absence of CMQ findings, we could consider item-level analyses. However, they also state that they do “not think it [is] necessary given the current status of the paper and its already rich set of analyses.” We have decided against these additional analyses for several reasons.

Firstly, the very purpose of items loading on the same factor is to form a psychometric scale, as single-item measures have inherent challenges, for instance, regarding reliability and validity. Analyzing single items in this context would be motivated by trying to find significant results despite the overall scale associations being non-significant, even though we have tested for associations no less than six times. The evidence clearly indicates that the CMQ does not predict conspiracy theory support online in this case, and efforts should not be made to artificially increase the number of tests to alter this conclusion.

Secondly, as the reviewer notes, the paper already contains a comprehensive set of analyses. The problematic approach of analyzing item-level responses whenever psychometric scales show no association would apply to all other measures that yield non-significant results.

Thirdly, due to multicollinearity, since items of the same scale are highly correlated by definition, we cannot test the influence of all items simultaneously. Thus, we would be unable to determine which item uniquely (i.e., controlling for all other items and the other scales) predicts conspiracy theory support.

The CMQ has become increasingly controversial, even prompting a special issue, which we partially cite in our papers. Our results contribute to this significant discourse by demonstrating that it may lack predictive validity, at least in this study setting.